# U³CF: Unbiased, Unconfounding, and Unified Causal Framework for Multi-Target Domain Adaptation

**Wenxu Wang** [1 2]  **Yeqiang Liu** [2]  **Rui Zhou** [3]  **Jing Wang** [4]  **Zhenbo Li** [2]  **Wenbo Gong** [5]

## Abstract

Multi-target domain adaptation (MTDA) trains a model using a labeled source domain and several unlabeled target domains, aiming to enhance performance across all targets. However, existing methods lack a principled causal formulation and often rely on empirical domain-invariance enforcement, which can bias adaptation across targets. To fill this gap, we propose the **U**nbiased, **U**nconfounding, and **U**nified **C**ausal **F**ramework (**U³CF**) for MTDA. To *unify* the alignment of multiple domains, we propose a prototype-driven alignment strategy that progressively updates prototypes by high-confidence target predictions, while the contrastive optimization objective jointly aligns target samples to semantic prototypes and preserves class discrimination. By formulating a structural causal model, we reveal that domain-invariant causal factors and domain-specific factors shape representations and labels, while the latter induce spurious label correlations across targets. Accordingly, U³CF achieves *unbiased* prediction by disentangling representations into invariant causal components and domain-specific confounders and applying conditional intervention to *block confounding* effects while preserving invariant semantics. To ensure precise disentanglement, we leverage mutual information theory to derive a principled criterion for feature separation. Extensive experiments on four benchmarks demonstrate that U³CF consistently outperforms leading methods.

---

[1]College of Computer Science and Technology, Ocean University of China, Qingdao, China [2]Department of Computer Engineering, College of Information and Electrical Engineering, China Agricultural University, Beijing, China [3]College of Systems Engineering, National University of Defense Technology, Changsha, China [4]Department of Electrical and Computer Engineering, University of Alberta, Edmonton, Canada [5]Microsoft Research Cambridge, UK. Correspondence to: Zhenbo Li <lizb@cau.edu.cn>.

*Proceedings of the 43rd International Conference on Machine Learning*, Seoul, South Korea. PMLR 306, 2026. Copyright 2026 by the author(s).

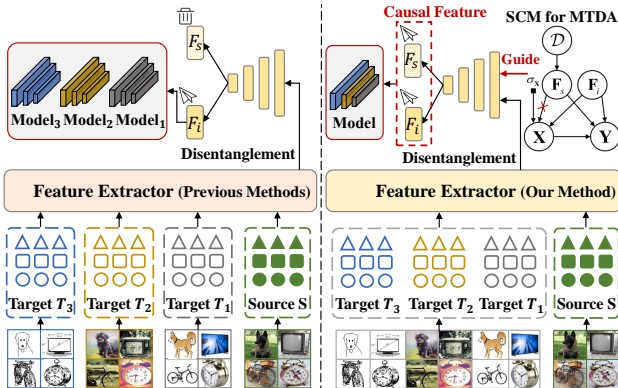

*Figure 1.* Prior STDA methods typically adapt a model to one target domain at a time, requiring re-adaptation for each new target. In contrast, our causally principled MTDA framework trains a single model that generalizes across multiple target domains.

## 1. Introduction

Deep neural networks often generalize poorly to out-of-distribution (OOD) test data when the independent and identically distributed (i.i.d.) assumption between training (source) and testing (target) distributions is violated (Chen et al., 2020). Since re-annotating data and retraining models for each new environment is costly, unsupervised domain adaptation (UDA) offers a practical alternative by mitigating domain shift and improving unlabeled-target performance.

Most UDA methods focus on adapting from a single labeled source domain to one unlabeled target domain, i.e., single-target domain adaptation (STDA). Representative STDA approaches are based on statistical alignment (Wang et al., 2023a; 2024; Long et al., 2019), adversarial training (Ganin & Lempitsky, 2015; Long et al., 2018; Bousmalis et al., 2017; Kang et al., 2018; Murez et al., 2018), and generative modeling (Liu et al., 2019; Wang et al., 2019; Chang et al., 2019; Russo et al., 2018). However, a key limitation is that an STDA model is typically specialized to the adapted target domain and must be re-adapted when deployed in new OOD environments, which hinders generalization across diverse targets (Fig. 1). Motivated by the abundance of unlabeled data in practice, recent work has moved toward multi-target domain adaptation (MTDA), which aims to adapt a single source model to multiple target domains and overcome the

limitations of STDA.

The MTDA paradigm aims to train a model on a labeled source domain and multiple unlabeled target domains, achieving strong performance across all targets. Existing MTDA studies remain relatively exploratory, with many methods adopting adversarial learning for adaptation (Roy et al., 2021; Xu et al., 2023; Gholami et al., 2020; Nguyen-Meidine et al., 2021; Chen et al., 2019; Huang et al., 2024). Most of them pursue domain invariance by aligning the source distribution with each target domain independently. However, several challenges persist in MTDA. First, discrepancy-based alignment methods (e.g., MMD) may degrade under heterogeneous and multiple domain shifts. Second, existing approaches lack a principled causal formulation to guide MTDA classification. Third, naively suppressing domain-specific features can hurt adaptation by ignoring essential confounding factors, which are crucial for achieving unbiased adaptation across diverse domains.

To address these challenges, we propose the **U**nbiased, **U**nconfounding, and **U**nified **C**ausal **F**ramework (**U**$^3$**CF**) for MTDA. U$^3$CF operates in two stages: (i) a progressive prototype-guided alignment strategy to reduce cross-domain divergence across multiple targets, and (ii) a causal adaptation formulation with feature disentanglement and conditional intervention under a structural causal model to mitigate confounding and enable unbiased adaptation.

Specifically, to handle heterogeneous shifts across multiple targets in MTDA, we propose a progressive class prototype-oriented cross-domain alignment strategy. We initialize the memory module with source features and iteratively expand it by incorporating high-confidence target samples, yielding globally representative class prototypes. We further adopt a contrastive objective that forms positive pairs between each target sample and its corresponding class prototype to promote cross-domain alignment. MTDA exhibits domain-specific factors due to distributional variations across domains. From a causal perspective, these factors may induce spurious label correlations and hinder generalization. To address this issue, we develop a domain-unbiased causal classifier. We first formulate a structural causal model for MTDA, in which domain-specific factors act as confounders. Leveraging mutual information theory, we derive a principled disentanglement criterion to separate domain-invariant causal features from domain-specific factors, avoiding ad-hoc invariance heuristics. Moreover, we model domain prototypes by updating $\mathcal{M}$ with domain-specific features, enabling explicit estimation of confounders. Finally, we perform conditional intervention to block confounding effects and reconstruct causal representations, achieving domain-unbiased classification across multiple target domains. The main contributions are summarized as follows:

• **Causal perspective**: We propose a novel structural causal

model (SCM) for MTDA and introduce a soft intervention (conditional intervention) strategy to alleviate the negative effects of domain-specific factors on prediction.

• **Theoretical perspective**: We provide principled guarantees, including (i) a contrastive lower bound on mutual information (Prop. 1); (ii) domain-unbiased prediction via conditional intervention (Prop. 2); and (iii) variational bounds for feature disentanglement and learning (Props. 3–4).

• **Methodological perspective**: We present U$^3$CF, which integrates (i) *Class Prototype Oriented Cross-Domain Alignment* to mitigate heterogeneous multi-domain shifts, (ii) *Mutual Information-based Feature Disentanglement Mechanism* for reliable representation separation, and (iii) *Feature Disentanglement-Based Domain-Unbiased Causal Classification* for cross-domain prediction in MTDA.

• **Experimental perspective**: Extensive experiments on standard benchmarks demonstrate that U$^3$CF consistently outperforms the leading baselines.

## 2. Methodology

**Notation.** Scalars, vectors, and matrices are denoted by $x$, $\mathbf{x}$, and $\mathbf{X}$, respectively. Variables in the structural causal model are written in non-bold (e.g., $X$), while their learned representations used for computation are bold (e.g., $\mathbf{x}_i$, $\mathbf{X}$).

In the MTDA paradigm, we consider a single fully labeled source domain, denoted by $\mathcal{D}_S = \{(\mathbf{x}_{si}, y_{si})\}_{i=1}^{n_s} = \{\mathbf{X}_s, \mathbf{y}_s\}$, which contains $n_s$ samples, along with multiple unlabeled target domains $\mathcal{D}_T = \{\mathcal{D}_{ti}\}_{i=1}^{T}$. Each target domain $\mathcal{D}_{ti} = \{\mathbf{x}_{tj}\}_{j=1}^{n_{ti}} = \mathbf{X}_{ti}$ comprises $n_{ti}$ samples. The total number of domains is $K_D = 1 + T$, and all domains share $K$ categories. Fig. 2 shows that U$^3$CF consists of two stages: cross-domain alignment and causal classification.

### 2.1. Progressive Class Prototype Oriented Cross-Domain Alignment

**Overview.** As shown in Fig. 2, we first train the memory bank module $\mathcal{M}$ on the source data and extract class prototypes from $\mathcal{M}$ for cross-domain alignment. During alignment, $\mathcal{M}$ and the prototypes are iteratively expanded and refined. After alignment, we select target samples with high pseudo-label confidence and incorporate them into the source domain, forming a high-confidence domain $\mathcal{D}_H$.

#### 2.1.1. MEMORY BANK INITIALIZATION AND CLASS PROTOTYPE LEARNING

**Memory bank initialization.** We first train the model on the source samples to construct the memory bank $\mathcal{M}$, which stores the feature matrix $\mathbf{F} \in \mathbb{R}^{n_s \times h}$ (where $h$ denotes the feature dimension), category labels $\mathbf{y}_c \in [1, \cdots, K]^{n_s \times 1}$, and domain labels $\mathbf{y}_d \in [1, \cdots, K_D]^{n_s \times 1}$.

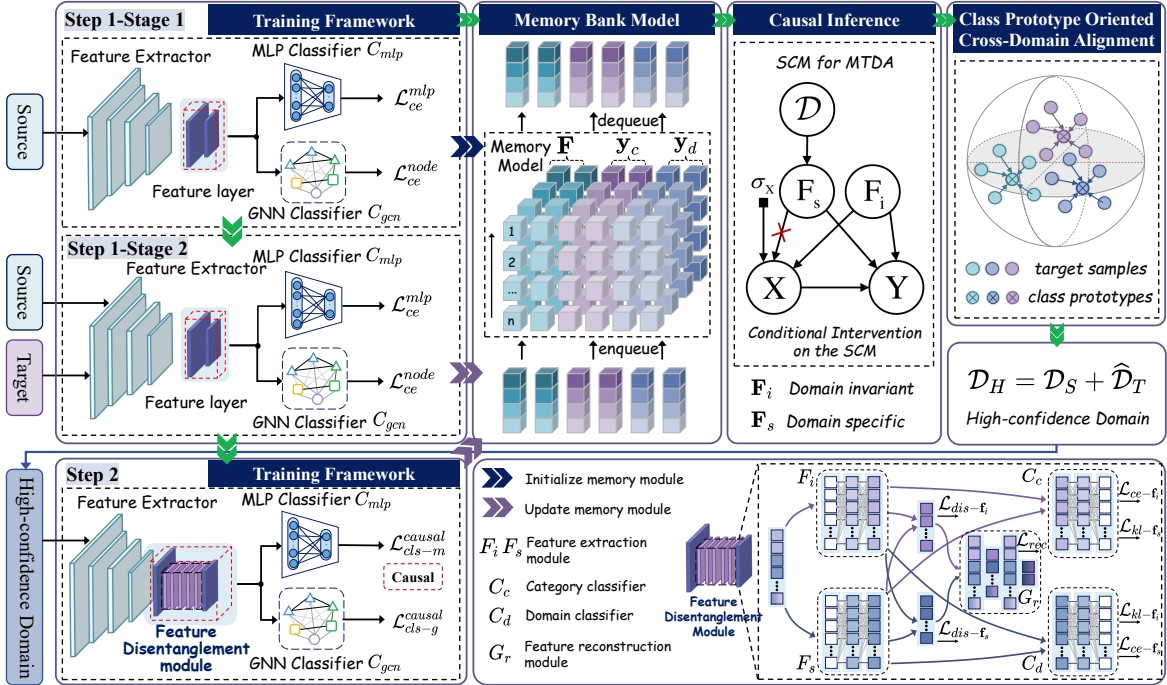

*Figure 2.* Overview of the proposed U³CF framework, which consists of two steps: cross-domain alignment (step 1) and causal classification (step 2). In stage 1 of step 1, the model is trained on the source domain and the memory module $\mathcal{M}$ is initialized. In stage 2 of step 1, cross-domain alignment is performed using both source and target domains, and the $\mathcal{M}$ is dynamically updated. In step 2, we incorporate a feature disentanglement module to separate domain-invariant features and domain-specific features, and then derive a domain-unbiased causal classifier via conditional intervention. The Feature Disentanglement Module (FDM) is used in Sec. 2.3.

Denote $\mathbf{z}_{si}$ as the feature vector of the source sample $\mathbf{x}_{si}$ extracted by the feature extraction network $G_\Theta$; $\mathbf{z}_{si}$ is used to initialize and update $\mathbf{F}$.

**Class prototype learning.** The class prototype $\mathbf{c}_s^k$ for the k-th class in the source domain is computed as $\mathbf{c}_s^k = \frac{1}{n_s^k}\sum_{i=1}^{n_s^k}\mathbf{z}_{si}$, where $n_s^k$ denotes the number of source samples of the k-th class in $\mathbf{F}$.

### 2.1.2. CLASS PROTOTYPE ORIENTED CROSS-DOMAIN ALIGNMENT

Motivated by contrastive learning, which encourages high similarity between positive pairs (van den Oord et al., 2018; He et al., 2020; Khosla et al., 2020), we propose a contrastive-inspired class prototype oriented cross-domain alignment loss. The proposed objective pulls each target sample $\mathbf{x}_{ti}$ toward its corresponding class prototype (treated as a positive), promoting consistent global structure alignment across domains. In addition, it leverages discriminative target features from the unlabeled target domains $\{\mathcal{D}_{ti}\}_{i=1}^T$ to construct a high-confidence sample set $\mathcal{D}_H$, which further improves the representativeness of the learned class prototypes. This design also facilitates effective alignment across multiple target domains and improves the quality of target pseudo labels.

Denote $\mathbf{z}_{ti}$ as the latent feature of the target sample $\mathbf{x}_{ti}$

extracted by the feature extraction network. The contrastive-inspired class prototype oriented cross-domain alignment loss can be expressed as:

$$\mathcal{L}_{ali} = -\sum_{i=1}^{n_t}\log\frac{f\left(\mathbf{z}_{ti}, \mathbf{c}_s^{k+}\right)}{\sum_{k=1}^K f\left(\mathbf{z}_{ti}, \mathbf{c}_s^k\right)} \quad (1)$$

**Connection to mutual information.** Eq. (1) is an InfoNCE-style objective over $K$ class prototypes (van den Oord et al., 2018). The following proposition states that minimizing $\mathcal{L}_{ali}$ maximizes a lower bound on the mutual information between target features and their same-class prototypes.

**Proposition 1** (Contrastive lower bound on mutual information). *Let $\mathcal{L}_{ali}(\mathbf{z}_{ti})$ denote the per-sample term in Eq. (1). For each $\mathbf{z}_{ti}$, the positive prototype $\mathbf{c}_s^{k+}$ is paired with $\mathbf{z}_{ti}$, while the remaining $K-1$ prototypes in the denominator act as negatives drawn independently of $\mathbf{z}_{ti}$, we have*

$$I\left(\mathbf{z}_{ti}; \mathbf{c}_s^{k+}\right) \geq \log K - \mathbb{E}[\mathcal{L}_{ali}(\mathbf{z}_{ti})]. \quad (2)$$

*Equivalently, minimizing $\mathcal{L}_{ali}$ maximizes a lower bound of $I\left(\mathbf{z}_{ti}; \mathbf{c}_s^{k+}\right)$ up to the additive constant $\log K$.*

**Proof sketch.** Interpreting Eq. (1) as a $K$-way contrastive

classification objective yields a mutual-information lower bound. See Appendix A.1 for details.

### 2.1.3. PROGRESSIVE CLASS PROTOTYPE UPDATE

During cross-domain alignment, we sequentially align source samples to each target domain, where the target-domain order is determined by the entropy (Roy et al., 2021) of each target domain. Target sample selection follows the maximum pseudo-label probability, with a threshold set to $\theta$. After each alignment step, high-confidence target samples $\widehat{\mathcal{D}}_T = \{\widehat{\mathcal{D}}_{ti}\}_{i=1}^T$ are merged into the source domain $\mathcal{D}_S$, yielding a new high-confidence domain $\mathcal{D}_H$ (i.e., $\mathcal{D}_H = \mathcal{D}_S + \widehat{\mathcal{D}}_T$). Meanwhile, we adopt a queue-like strategy to progressively expand the memory module $\mathcal{M}$ by integrating sample and label information from $\mathcal{D}_H$, which comprises $n_h$ samples.

With each alignment step, class prototypes are progressively updated using the continuously refreshed memory module $\mathcal{M}$. These refined prototypes incorporate not only discriminative source features but also high-confidence target features, thereby exploiting information from the unlabeled target domains and improving alignment accuracy in MTDA.

### 2.2. Structural Causal Model for MTDA

**Overview.** As illustrated in Fig. 2, the second step introduces a novel SCM tailored to MTDA to construct a causal classifier. Specifically, using $\mathcal{D}_H$ as training data, we train the neural network with mutual information to disentangle domain-invariant features $\mathbf{F}_i$ and domain-specific features $\mathbf{F}_s$ for each image. In addition, $\mathbf{F}_s$ and the domain labels stored in $\mathcal{M}$ are updated to derive domain prototypes. These domain prototypes are then treated as confounders in the SCM, and conditional interventions are applied to obtain an unbiased causal classifier for MTDA.

In this stage, all samples in $\mathcal{D}_H$, including high-confidence target samples $\widehat{\mathcal{D}}_T$ with pseudo-labels, are used for model training. By combining feature disentanglement with SCM formulation (Pearl, 2000; Gong et al., 2023; Chen et al., 2022; Wang et al., 2025) for MTDA, the proposed approach ultimately enables domain-unbiased causal classification.

We propose a causal framework for MTDA and formulate the SCM. As shown in Fig. 3(a), $\mathcal{D}$ denotes the domain and $y_d \in [1, K_D]$ is its domain label. $F_s$ denotes the domain-specific features, while $F_i$ denotes the domain-invariant features. $X$ represents the input data and $Y$ denotes the category label.

$X \to Y$ represents the causal relationship to be learned: predicting the category label $Y$ from the input feature $\mathbf{X}$ (i.e., $\mathbf{X} \in \mathcal{D}_H$). $\mathcal{D} \to F_s$ indicates that each domain exhibits its own domain-specific features $\mathbf{F}_s$, such as acquisition conditions, backgrounds, illumination, and other latent factors.

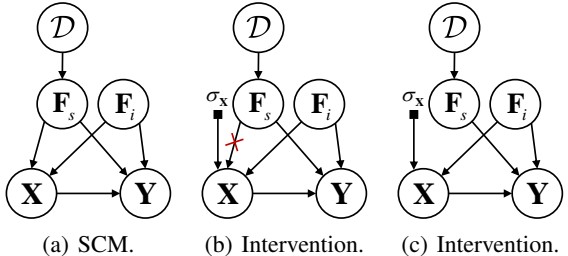

(a) SCM.    (b) Intervention.    (c) Intervention.

*Figure 3.* The SCM for the MTDA paradigm. ((a) SCM for MTDA. (b) and (c) Conditional intervention in the SCM.)

In the fork structure $X \leftarrow F_s \to Y$, domain-specific factors induce distribution shifts across domains, influencing both the generation of $X$ and the prediction of $Y$; hence, $F_s$ acts as a confounder in the SCM. Similarly, $X \leftarrow F_i \to Y$ captures that domain-invariant features affect both $X$ and $Y$, and thus $F_i$ also serves as a confounder in the SCM.

The domain-specific factor $F_s$ confounds the relationship between $X$ and $Y$, which can bias the observational prediction $P(Y \mid X)$ through the backdoor path $X \leftarrow F_s \to Y$ in Fig. 3(b). To mitigate this confounding effect while preserving the predictive role of the invariant factor $F_i$, we consider a *conditional intervention* that replaces the data-generating mechanism of $X$ with a deterministic policy driven by $F_i$, i.e., $\sigma_X = do(X = g(F_i))$ (Correa & Bareinboim, 2020). This "soft" intervention keeps the rest of the structural causal model unchanged and yields an interventional predictor that averages out the influence of $F_s$. In our implementation, the policy output $g(\mathbf{f}_i)$ is implicitly parameterized by the feature reconstruction module $G_r$ in Fig. 2.

> **Proposition 2** (Conditional intervention with domain averaging). *Consider the SCM in Fig. 3(b)–(c), where $F_s$ is a domain-specific confounder affecting both $X$ and $Y$, and $F_i$ is a domain-invariant factor. Define the conditional intervention $\sigma_X = do(X = g(F_i))$, which replaces the mechanism of $X$ by the deterministic policy $g(F_i)$, i.e., $P(\mathbf{x} \mid F_i = \mathbf{f}_i, F_s = \mathbf{f}_s; \sigma_X) = \delta_{\mathbf{x}=g(\mathbf{f}_i)}$. Then for any $\mathbf{f}_i$, the intervention predictive distribution satisfies*
>
> $$P(Y \mid F_i = \mathbf{f}_i; \sigma_X) =$$
> $$\sum_{F_s} P(Y \mid F_s, F_i = \mathbf{f}_i, X = g(\mathbf{f}_i)) \, P(F_s) \quad (3)$$
>
> *Moreover, if $F_s$ is approximated by $K_D$ domain prototypes $\{\widehat{\mathbf{f}}_{sj}\}_{j=1}^{K_D}$ with uniform weights $P(F_s = \widehat{\mathbf{f}}_{sj}) = \frac{1}{K_D}$, then*
>
> $$P(Y \mid F_i = \mathbf{f}_i; \sigma_X) =$$
> $$\frac{1}{K_D} \sum_{j=1}^{K_D} P\left(Y \mid F_s = \widehat{\mathbf{f}}_{sj}, F_i = \mathbf{f}_i, X = g(\mathbf{f}_i)\right) \quad (4)$$

**Notation.** $X, Y, F_s, F_i$ denote random variables in the SCM; $\mathbf{f}_i$ and $\widehat{\mathbf{f}}_{sj}$ denote their (vector) realizations; $\delta_{\mathbf{x}=g(\mathbf{f}_i)}$ is a Dirac/Kronecker delta indicating the deterministic assignment under $\sigma_X$.

**Proof sketch.** Full proof is provided in Appendix A.2.

Although target domains are unlabeled, their domain identities are known in MTDA, which enables us to estimate domain-specific confounders (via domain prototypes) and instantiate the domain-averaged interventional predictor in Eq. (4).

### 2.3. Feature Disentanglement via Mutual Information

2.3.1. THE DISENTANGLEMENT OF $\mathbf{f}_i$ AND $\mathbf{f}_s$.

**Motivation.** The domain-specific confounder $F_s$ is typically entangled in the observed feature $X$ (with realizations $\mathbf{x}$), which makes it difficult to explicitly instantiate the confounder required by Eq. (4). To make Eq. (4) operational, we introduce a feature disentanglement module that decomposes each representation $\mathbf{x}$ into a domain-invariant feature $\mathbf{f}_i$ and a domain-specific feature $\mathbf{f}_s$ (in our implementation, $\mathbf{f}_i, \mathbf{f}_s \in \mathbb{R}^{256 \times 1}$). Across domains, the collection of domain-specific factors $\mathbf{f}_s$ is further summarized into $K_D$ domain prototypes $\{\widehat{\mathbf{f}}_{sj}\}_{j=1}^{K_D}$, which provide a practical discretization of $F_s$ for estimating the domain-averaged intervention in Eq. (4). Concretely, we maintain $K_D$ instances of $\widehat{\mathbf{f}}_{sj}$ and compute the above quantities over $n_h$ feature instances $\mathbf{x}$.

**Disentanglement principle.** To make the domain-averaged intervention in Eq. (4) operational, we explicitly instantiate the confounder through a feature decomposition $\mathbf{x} \mapsto (\mathbf{f}_i, \mathbf{f}_s)$, where $\mathbf{f}_i$ captures domain-invariant factors and $\mathbf{f}_s$ captures domain-specific factors. We encourage $\mathbf{f}_i$ and $\mathbf{f}_s$ to represent complementary information by minimizing their mutual information (MI) $I(\mathbf{f}_i; \mathbf{f}_s)$. Using the interaction-information identity (McGill, 1954; Hwang et al., 2020), we have

$$I(\mathbf{f}_i; \mathbf{f}_s) = I(\mathbf{f}_i; \mathbf{x}) - I(\mathbf{f}_i; \mathbf{x} \mid \mathbf{f}_s) + I(\mathbf{f}_i; \mathbf{f}_s \mid \mathbf{x}) \quad (5)$$

Following standard variational disentanglement modeling, we use an inference factorization in which $\mathbf{f}_i$ does not additionally depend on $\mathbf{f}_s$ once $\mathbf{x}$ is given. We use a factorized inference model for the two components, i.e., $p(\mathbf{f}_i, \mathbf{f}_s | \mathbf{x}) = p(\mathbf{f}_i | \mathbf{x}) p(\mathbf{f}_s | \mathbf{x})$, which implies $I(\mathbf{f}_i; \mathbf{f}_s | \mathbf{x}) = 0$. Consequently, $I(\mathbf{f}_i; \mathbf{f}_s)$ admits the decomposition

$$I(\mathbf{f}_i; \mathbf{f}_s) = I(\mathbf{x}; \mathbf{f}_i) + I(\mathbf{x}; \mathbf{f}_s) - I(\mathbf{x}; \mathbf{f}_i, \mathbf{f}_s) \quad (6)$$

Eq. (6) reveals a natural trade-off: we suppress the information that $\mathbf{f}_i$ and $\mathbf{f}_s$ individually retain about $\mathbf{x}$, while requiring their joint representation to remain sufficiently informative to reconstruct $\mathbf{x}$.

**Variational surrogate with information bounds.** Directly optimizing the mutual information (MI) terms in Eq. (6)

is generally intractable. We therefore introduce reference marginals $q(\mathbf{f}_i)$ and $q(\mathbf{f}_s)$ (standard Gaussians $\mathcal{N}(\mathbf{0}, \mathbf{I})$ in practice) and a decoder $q(\mathbf{x} | \mathbf{f}_i, \mathbf{f}_s)$. Let $p(\mathbf{f}_i | \mathbf{x})$ and $p(\mathbf{f}_s | \mathbf{x})$ denote the encoder-induced posteriors. We define two KL regularizers and a reconstruction loss:

$$\mathcal{L}_{dis-\mathbf{f}_i} \triangleq \mathbb{E}_{p(\mathbf{x})} \big[ D_{\text{KL}}\big( p(\mathbf{f}_i \mid \mathbf{x}) \| q(\mathbf{f}_i) \big) \big] \quad (7)$$

$$\mathcal{L}_{dis-\mathbf{f}_s} \triangleq \mathbb{E}_{p(\mathbf{x})} \big[ D_{\text{KL}}\big( p(\mathbf{f}_s \mid \mathbf{x}) \| q(\mathbf{f}_s) \big) \big] \quad (8)$$

$$\mathcal{L}_{rec} \triangleq - \mathbb{E}_{p(\mathbf{x})} \Big[ \mathbb{E}_{p(\mathbf{f}_i|\mathbf{x}) \, p(\mathbf{f}_s|\mathbf{x})} \big[ \log q(\mathbf{x} \mid \mathbf{f}_i, \mathbf{f}_s) \big] \Big] \quad (9)$$

We minimize the combined disentanglement objective

$$\mathcal{L}_{dis} \triangleq \mathcal{L}_{dis-\mathbf{f}_i} + \mathcal{L}_{dis-\mathbf{f}_s} + \mathcal{L}_{rec} \quad (10)$$

> **Proposition 3** (Variational bound for invariant–specific feature disentanglement). *Under the factorized encoder posteriors $p(\mathbf{f}_i, \mathbf{f}_s \mid \mathbf{x}) = p(\mathbf{f}_i \mid \mathbf{x}) p(\mathbf{f}_s \mid \mathbf{x})$, consider the surrogate objective $\mathcal{L}_{dis}$ in Eq. (10), with $\mathcal{L}_{dis-\mathbf{f}_i}$, $\mathcal{L}_{dis-\mathbf{f}_s}$ and $\mathcal{L}_{rec}$ defined in Eqs. (7)–(9). Then the following bound holds:*
>
> $$I(\mathbf{f}_i; \mathbf{f}_s) \leq \mathcal{L}_{dis} - H(\mathbf{x}) \quad (11)$$
>
> *where $H(\mathbf{x})$ depends only on the data distribution and is constant w.r.t. the model parameters. Therefore, minimizing $\mathcal{L}_{dis}$ minimizes an upper bound on $I(\mathbf{f}_i; \mathbf{f}_s)$.*

**Proof sketch.** Full proof is provided in Appendix A.3.

2.3.2. THE LEARNING OF $\mathbf{f}_i$ AND $\mathbf{f}_s$.

To ensure that the disentangled representations indeed exhibit the desired properties, we further impose *label-guided* constraints on $\mathbf{f}_i$ and $\mathbf{f}_s$. Concretely, we leverage two supervision signals: the category label $\mathbf{y}_c$ (available in labeled source domains) and the domain label $\mathbf{y}_d$ (available for all domains under the MTDA setting). We introduce two lightweight prediction heads to parameterize the corresponding predictive distributions: a category classifier $C_c$ that outputs $q_c(\mathbf{y}_c | \mathbf{f})$ over $K$ classes, and a domain classifier $C_d$ that outputs $q_d(\mathbf{y}_d | \mathbf{f})$ over $K_D$ domains. Intuitively, the domain-invariant representation $\mathbf{f}_i$ should be *category-relevant* yet *domain-irrelevant*, whereas the domain-specific representation $\mathbf{f}_s$ should be *domain-relevant* yet *category-irrelevant*. We formalize this principle using mutual-information objectives:

$$\max \ I(\mathbf{f}_i; \mathbf{y}_c); \quad \min \ I(\mathbf{f}_i; \mathbf{y}_d)$$
$$\max \ I(\mathbf{f}_s; \mathbf{y}_d); \quad \min \ I(\mathbf{f}_s; \mathbf{y}_c) \quad (12)$$

Below we show that these objectives admit tractable variational surrogates that lead exactly to our four losses. Specifically, Proposition 4 provides a lower bound for MI maximization (label relevance) and an upper bound for MI minimization (label irrelevance), yielding cross-entropy and KL-to-uniform losses, respectively.

**Proposition 4** (Variational bounds for label relevance and label irrelevance). *Label relevance.* *Let* $\mathbf{y}$ *be a discrete label and* $q(\mathbf{y} \mid \mathbf{f})$ *be any variational classifier. Then*

$$I(\mathbf{f}; \mathbf{y}) \geq H(\mathbf{y}) + \mathbb{E}_{p(\mathbf{f}, \mathbf{y})}\big[\log q(\mathbf{y} \mid \mathbf{f})\big]$$
$$= H(\mathbf{y}) - \mathcal{L}_{\text{cross-entropy}} \quad (13)$$

*Since* $H(\mathbf{y})$ *is constant w.r.t. the model parameters, minimizing* $\mathcal{L}_{\text{cross-entropy}}$ *maximizes the above lower bound and thus increases* $I(\mathbf{f}; \mathbf{y})$.

*Label irrelevance.* *Moreover, for any reference marginal* $r(\mathbf{y})$, *we have*

$$I(\mathbf{f}; \mathbf{y}) = \mathbb{E}_{p(\mathbf{f})}\big[D_{\mathrm{KL}}\big(p(\mathbf{y} \mid \mathbf{f}) \,\|\, r(\mathbf{y})\big)\big]$$
$$- D_{\mathrm{KL}}\big(p(\mathbf{y}) \,\|\, r(\mathbf{y})\big)$$
$$\leq \mathbb{E}_{p(\mathbf{f})}\big[D_{\mathrm{KL}}\big(p(\mathbf{y} \mid \mathbf{f}) \,\|\, r(\mathbf{y})\big)\big] \quad (14)$$

*In particular, choosing* $r$ *as the uniform distribution yields an upper bound that encourages label-agnostic features.*

**Label relevance via cross-entropy (MI maximization).** Applying Eq. (13) with $(\mathbf{f}, \mathbf{y}) = (\mathbf{f}_i, \mathbf{y}_c)$ and the classifier $q(\mathbf{y}_c | \mathbf{f}_i)$ parameterized by $C_c$, maximizing $I(\mathbf{f}_i; \mathbf{y}_c)$ is equivalent (up to the constant $H(\mathbf{y}_c)$) to minimizing the cross-entropy loss:

$$\mathcal{L}_{ce-\mathbf{f}_i} \triangleq -\mathbb{E}_{p(\mathbf{f}_i, \mathbf{y}_c)}\big[\log q(\mathbf{y}_c \mid \mathbf{f}_i)\big] \quad (15)$$

Similarly, applying Eq. (13) to $(\mathbf{f}, \mathbf{y}) = (\mathbf{f}_s, \mathbf{y}_d)$ with $q(\mathbf{y}_d | \mathbf{f}_s)$ parameterized by $C_d$ gives

$$\mathcal{L}_{ce-\mathbf{f}_s} \triangleq -\mathbb{E}_{p(\mathbf{f}_s, \mathbf{y}_d)}\big[\log q(\mathbf{y}_d \mid \mathbf{f}_s)\big] \quad (16)$$

Both losses encourage the corresponding features to be predictive of the intended labels.

**Label irrelevance via KL-to-uniform (MI minimization).** To enforce domain irrelevance for $\mathbf{f}_i$, we minimize $I(\mathbf{f}_i; \mathbf{y}_d)$. Using Eq. (14) with reference marginal $r(\mathbf{y}_d) = u_d(\mathbf{y}_d) = \frac{1}{K_D}$ (uniform over $K_D$ domains), we obtain an upper bound in terms of the KL divergence between the domain posterior and the uniform distribution. This yields the tractable regularizer

$$\mathcal{L}_{kl-\mathbf{f}_i} \triangleq \mathbb{E}_{p(\mathbf{f}_i)}\big[D_{\mathrm{KL}}\big(p(\mathbf{y}_d \mid \mathbf{f}_i) \,\|\, u_d(\mathbf{y}_d)\big)\big] \quad (17)$$

Minimizing $\mathcal{L}_{kl-\mathbf{f}_i}$ drives $p(\mathbf{y}_d | \mathbf{f}_i)$ towards uniform, thereby suppressing domain-discriminative information in $\mathbf{f}_i$.

Similarly, to enforce category invariance for $\mathbf{f}_s$, we choose $r(\mathbf{y}_c) = u_c(\mathbf{y}_c) = \frac{1}{K}$ (uniform on $K$ categories) and define

$$\mathcal{L}_{kl-\mathbf{f}_s} \triangleq \mathbb{E}_{p(\mathbf{f}_s)}\big[D_{\mathrm{KL}}\big(p(\mathbf{y}_c \mid \mathbf{f}_s) \,\|\, u_c(\mathbf{y}_c)\big)\big] \quad (18)$$

Since $H(\mathbf{y}) \leq \log|\mathcal{Y}|$, the uniform reference provides a principled label-agnostic constraint. Together, Eqs. (15)–(18) implement the MI objectives in Eq. (12) with tractable variational surrogates.

## 2.4. Feature Disentanglement-Based Domain Unbiased Causal Classification

Eq. (4) states that the desired interventional prediction under $\sigma_X = do\big(X = g(F_i)\big)$ is a *domain-averaged* conditional: for a given $\mathbf{f}_i$, we predict by marginalizing the domain-specific confounder $F_s$ with weights $P(F_s)$. To implement this idea with finite samples, we approximate $F_s$ by $K_D$ domain prototypes $\{\widehat{\mathbf{f}}_{sj}\}_{j=1}^{K_D}$ and use the uniform prior $P(F_s = \widehat{\mathbf{f}}_{sj}) = \frac{1}{K_D}$, which yields an *unbiased* (domain-agnostic) intervention: each domain contributes equally to the prediction regardless of its frequency in the mini-batch or dataset.

**Domain-unbiased causal losses.** Given $\mathbf{f}_i$ and each prototype $\widehat{\mathbf{f}}_{sd}$, we form a causal representation $G_r(\mathbf{f}_i, \widehat{\mathbf{f}}_{sd})$ and *uniformly average* the logits over all $K_D$ prototypes to approximate the domain-averaged intervention in Eq. (4). The resulting causal classification losses are

$$\mathcal{L}_{cls-m}^{causal} = \mathcal{L}_{ce}\Big(\frac{1}{K_D}\sum_{d=1}^{K_D} C_{mlp}(G_r(\mathbf{f}_i, \widehat{\mathbf{f}}_{sd})), \mathbf{y}_c\Big) \quad (19)$$

$$\mathcal{L}_{cls-g}^{causal} = \mathcal{L}_{ce}\Big(\frac{1}{K_D}\sum_{d=1}^{K_D} C_{gcn}(G_r(\mathbf{f}_i, \widehat{\mathbf{f}}_{sd})), \mathbf{y}_c\Big) \quad (20)$$

where $\mathcal{L}_{ce}$ denotes the cross-entropy loss. Crucially, the uniform averaging $\frac{1}{K_D}\sum_{d=1}^{K_D}(\cdot)$ makes the classifier *domain-unbiased*: it enforces equal confounder representation across domains, preventing the prediction from being dominated by domain-specific spurious correlations.

## 2.5. Overall Formulation

The proposed method shares the same setting as method D-CGCT (Roy et al., 2021) and both utilize two classifiers: an MLP classifier $C_{mlp}$ and a GCN classifier $C_{gcn}$. For fair and consistent comparison with prior graph-based baselines, we keep these baseline loss definitions $(\mathcal{L}_{ce}^{node}, \mathcal{L}_{bce}^{edge})$ unchanged.

**Training.** Training proceeds in two steps: *cross-domain alignment* and *causal classification*.

**Step 1 (cross-domain alignment).** Stage 1, Initialize $\mathcal{M}$. The training loss is $\mathcal{L}_{train} = \mathcal{L}_{ce}^{mlp} + \mathcal{L}_{bce}^{edge} + \lambda_{node}\mathcal{L}_{ce}^{node}$. Stage 2, Cross-domain alignment. The training loss is $\mathcal{L}_{train} = \mathcal{L}_{ce}^{mlp} + \mathcal{L}_{bce}^{edge} + \lambda_{node}\mathcal{L}_{ce}^{node} + \lambda_{ali}\mathcal{L}_{ali}$.

**Step 2 (causal classification).** We optimize the domain-unbiased causal classification losses together with disentanglement and label-guided supervision: $\mathcal{L}_{train} = \mathcal{L}_{cls-m}^{causal} + \mathcal{L}_{bce}^{edge} + \lambda_{node}\mathcal{L}_{cls-g}^{causal} + \mathcal{L}_{dis} + \lambda_{ce}(\mathcal{L}_{ce-\mathbf{f}_i} + \mathcal{L}_{ce-\mathbf{f}_s}) + \lambda_{kl}(\mathcal{L}_{kl-\mathbf{f}_i} + \mathcal{L}_{kl-\mathbf{f}_s})$, where $\mathcal{L}_{dis} = \lambda_{dis}(\mathcal{L}_{dis-\mathbf{f}_i} + \mathcal{L}_{dis-\mathbf{f}_s}) + \lambda_{rec}\mathcal{L}_{rec}$.

**Testing.** In the testing phase, the final prediction category $\mathbf{y}_{gcn}^{causal}$ is calculated by the GCN classifier $C_{gcn}$.

$$\mathbf{y}_{gcn}^{causal} = \arg\max\left(\frac{1}{K_D}\sum_{d=1}^{K_D} C_{gcn}(G_r(\mathbf{f}_i, \widehat{\mathbf{f}}_{sd}))\right) \quad (21)$$

where the $C_{gcn}$ outputs the softmax probabilities according to Eq. (4). The procedure is summarized in Algorithm 1.

## 3. Experimental Results and Discussion

### 3.1. Experimental Setup

We conduct experiments on four domain adaptation benchmarks, including *Office-Home*, *MTRS*, *Office-31*, and *DomainNet*. Additional details on the *baselines* and *experimental setup* are provided in Appendix D.

### 3.2. Experimental Results in the MTDA Setting

*Table 1.* Performance evaluation (%) against SOTA methods on **Office-Home**, utilizing ResNet-50 as the backbone. Best (Bold), second best (Underline).

| Setting | Model | Art | Clipart | Product | Real | **Avg** |
|---|---|---|---|---|---|---|
| Single-Target | JAN | 58.7 | 57.0 | 53.1 | 64.3 | 58.3 |
| | CDAN | 64.2 | 62.9 | 59.9 | 68.1 | 63.8 |
| | CGCT | 67.9 | 68.7 | 62.3 | 70.7 | 67.4 |
| | MEDM | 71.1 | 70.2 | 65.6 | 71.2 | 69.5 |
| | ODJID | 68.7 | 68.9 | 64.2 | 68.2 | 67.5 |
| Target-Combined | CDAN | 59.5 | 61.0 | 54.7 | 62.9 | 59.5 |
| | AMEAN | 64.3 | 65.5 | 59.5 | 66.7 | 64.0 |
| | CGCT | 67.4 | 68.1 | 61.6 | 68.7 | 66.5 |
| | DML | 67.0 | 70.6 | 62.9 | 69.0 | 67.4 |
| | MCDA | _71.7_ | _72.8_ | 68.0 | _71.7_ | _71.1_ |
| Multi-Target | MT-MTDA | 64.6 | 66.4 | 59.2 | 67.1 | 64.3 |
| | DCL | 63.0 | 66.3 | 60.0 | 67.0 | 64.1 |
| | D-CGCT | 70.5 | 71.6 | 66.0 | 71.2 | 69.8 |
| | DML | 67.4 | 70.2 | 64.2 | 70.0 | 68.0 |
| | CAIC | 69.3 | 69.4 | _68.3_ | 71.4 | 69.6 |
| | U³CF(Ours) | **74.2** | **75.3** | **69.7** | **73.9** | **73.3** |

*Table 2.* Performance evaluation (%) against SOTA methods on **MTRS**, utilizing ResNet-50 as the backbone. Best (Bold), second best (Underline).

| Setting | Model | A | N | P | R | U | **Avg** |
|---|---|---|---|---|---|---|---|
| Single-Target | JAN | 72.0 | 67.0 | 59.9 | 62.4 | 62.9 | 64.8 |
| | CDAN | 80.3 | 72.9 | 52.7 | 67.8 | 63.3 | 67.4 |
| | TADA | 81.3 | 72.3 | 55.0 | 64.0 | 51.8 | 64.9 |
| Target-Combined | CDAN | 71.4 | 69.4 | 45.8 | 64.1 | 58.7 | 61.9 |
| | TADA | 67.1 | 70.0 | 45.4 | 47.0 | 53.5 | 56.6 |
| | CGCT | 93.0 | _93.7_ | 67.1 | _90.3_ | 84.2 | 85.7 |
| | MCDA | 90.7 | 78.4 | 69.4 | 72.2 | 84.4 | 79.0 |
| Multi-Target | D-CGCT | _93.4_ | 91.7 | _73.2_ | 87.8 | _84.7_ | _86.2_ |
| | U³CF(Ours) | **93.8** | **94.9** | **74.7** | **91.2** | **85.7** | **88.1** |

As shown in Tables 1 to 4, we evaluate the proposed U³CF method against representative DA approaches across four benchmark datasets: Office-Home, MTRS, Office-31, and DomainNet. Key insights from these experiments are detailed as follows.

*Table 3.* Performance evaluation (%) against SOTA methods on **Office-31**, utilizing ResNet-50 as the backbone. Best (Bold), second best (Underline).

| Setting | Model | Amazon | DSLR | Webcam | **Avg** |
|---|---|---|---|---|---|
| Single-Target | JAN | 85.0 | 83.0 | 85.6 | 84.3 |
| | CDAN | 91.4 | 84.1 | 84.0 | 86.6 |
| | CGCT | 89.6 | 85.5 | 87.6 | 87.6 |
| | MEDM | 93.4 | 86.5 | 87.7 | 89.2 |
| | ODJID | 91.7 | 85.4 | 86.2 | 87.8 |
| Target-Combined | CDAN | 93.6 | 80.5 | 81.3 | 85.1 |
| | AMEAN | 90.1 | 77.0 | 73.4 | 80.2 |
| | CGCT | _93.9_ | 85.1 | 85.6 | 88.2 |
| | DML | 94.9 | 85.3 | 86.2 | 88.8 |
| | MCDA | 92.4 | **87.7** | **88.8** | _89.6_ |
| Multi-Target | MT-MTDA | 87.9 | 83.7 | 84.0 | 85.2 |
| | DCL | 92.6 | 82.5 | 84.7 | 86.6 |
| | D-CGCT | 93.4 | 86.0 | 87.1 | 88.8 |
| | DML | 96.8 | 85.3 | 86.1 | 89.4 |
| | CoNMix | 92.4 | 81.8 | 80.4 | 84.9 |
| | CAIC | 93.6 | 85.8 | 87.9 | 89.0 |
| | U³CF(Ours) | **94.3** | _87.6_ | _88.6_ | **90.2** |

*Table 4.* Performance evaluation (%) against SOTA methods on **DomainNet**, utilizing ResNet-101 as the backbone. Best (Bold), second best (Underline).

| Setting | Model | Cli. | Inf. | Pai. | Qui. | Rea. | Ske. | **Avg** |
|---|---|---|---|---|---|---|---|---|
| Target-Combined | SE | 21.3 | 8.5 | 14.5 | 13.8 | 16.0 | 19.7 | 15.6 |
| | MCD | 25.1 | 19.1 | 27.0 | 10.4 | 20.2 | 22.5 | 20.7 |
| | DADA | 26.1 | 20.0 | 26.5 | 12.9 | 20.7 | 22.8 | 21.5 |
| | CDAN | 31.6 | 27.1 | 31.8 | 12.5 | 33.2 | 35.8 | 28.7 |
| | MCC | 33.6 | 30.0 | 32.4 | 13.5 | 28.0 | 35.3 | 28.8 |
| | CGCT | 36.1 | 33.3 | 35.0 | 10.0 | 39.6 | 39.7 | 32.3 |
| | MCDA | _37.5_ | **37.3** | 36.6 | 17.8 | 36.1 | _41.4_ | _34.5_ |
| Multi-Target | DCL | 35.1 | 31.4 | 37.0 | _20.5_ | 35.4 | 41.0 | 33.4 |
| | D-CGCT | 37.0 | 32.2 | 37.3 | 19.3 | _39.8_ | 40.8 | 34.4 |
| | DML | 32.0 | 25.4 | 29.4 | 12.7 | 31.5 | 36.4 | 27.9 |
| | CoNMix | **41.8** | 29.2 | **39.9** | 17.5 | 32.7 | 41.2 | 33.7 |
| | U³CF(Ours) | 37.1 | _34.7_ | _39.2_ | **20.6** | **41.0** | **42.6** | **35.9** |

The results consistently demonstrate the superior average accuracy of U³CF across all datasets compared to other methods. Specifically, on Office-Home, U³CF achieves the best performance on all four MTDA tasks, surpassing the second-best method, MCDA, by 2.2% in average accuracy. While MCDA performs strongly due to its multi-level cross-domain alignment and emphasis on category importance, U³CF further incorporates domain labels, enabling the learning of domain-specific features that are essential for building a domain-unbiased MTDA model. On MTRS, U³CF remains the top performer on all five MTDA tasks, outperforming D-CGCT by 1.9% in average accuracy. On Office-31, U³CF achieves the best average performance, exceeding MCDA by 0.6% in average accuracy. On the highly challenging DomainNet benchmark, U³CF attains the best results on three out of six MTDA tasks and yields the highest average accuracy, outperforming MCDA by 1.4% and D-CGCT by 1.5%. Unlike D-CGCT, U³CF leverages domain-

specific features as confounders to enable domain-unbiased causal classification, further validating its effectiveness for MTDA.

The experimental results indicate that U³CF tends to benefit more as the number of target domains increases. Intuitively, each additional domain introduces distinct characteristics that can amplify the adverse influence of domain-specific features on model predictions. By constructing the SCM specifically for MTDA and applying conditional interventions, U³CF effectively counteracts the impact of domain-specific features, enabling domain-unbiased classification and improving performance on multi-domain datasets.

### 3.3. Experimental Results in the MSMTDA Setting

*Table 5.* Performance evaluation (%) compared with SOTA methods on **Office-Home** in the MSMTDA setting, using ResNet-50 as the backbone. Best results are in bold, and second-best results are underlined.

| Source | Rw+Pr | Cl+Rw | Pr+Cl | Rw+Ar | Ar+Pr | Cl+Ar | Avg. |
| Target | Ar+Cl | Ar+Pr | Ar+Rw | Cl+Pr | Cl+Rw | Pr+Rw | |
|---|---|---|---|---|---|---|---|
| MAN | 68.4 | 77.3 | 72.1 | 72.9 | 71.5 | 77.1 | 73.2 |
| IPCA | 65.6 | 79.9 | 75.2 | 70.1 | 71.8 | 79.3 | 73.7 |
| U³CF | **69.8** | **82.8** | **79.9** | **75.4** | **75.1** | **85.5** | **78.1** |

As shown in Table 5, U³CF achieves the best results across all six transfer tasks in the multi-source multi-target domain adaptation (MSMTDA) setting. It improves the average accuracy from 73.7% to 78.1% over the strongest baseline IPCA, with a gain of 4.4 percentage points. These results indicate that U³CF remains effective in the more challenging MSMTDA setting with multiple source and target domains.

*Table 6.* Ablations on **MTRS**. (*S1-s1*: Step 1-Stage 1; *S1-s2*: Step 1-Stage 2; *S2*: Step 2.)

| Model | S1-s1 | S1-s2 | S2 | MTRS | | | | | |
| | | | | A | N | P | R | U | **Avg(%)** |
|---|---|---|---|---|---|---|---|---|---|
| U³CF-1 | ✓ | | | 87.6 | 87.2 | 58.1 | 65.1 | 73.2 | 74.2 |
| U³CF-2 | | ✓ | | 91.9 | 92.0 | 69.9 | 85.7 | 82.1 | 84.3 |
| U³CF-3 | ✓ | ✓ | | 92.2 | 93.3 | 73.5 | 89.1 | 84.1 | 86.4 |
| U³CF | ✓ | ✓ | ✓ | **93.8** | **94.9** | **74.7** | **91.2** | **85.7** | **88.1** |

### 3.4. Ablation Study

We conduct an ablation study to examine the contribution of each component in U³CF. As described in Section 2.5, U³CF comprises two main steps: cross-domain alignment (step 1) and causal classification (step 2). Step 1 further contains two stages: stage 1 initializes the memory module $\mathcal{M}$, and stage 2 updates $\mathcal{M}$ while performing cross-domain alignment. For clarity, U³CF-1 corresponds to step 1-stage 1, U³CF-2 corresponds to step 1-stage 2, and U³CF-3 denotes the full step 1. The proposed U³CF integrates both

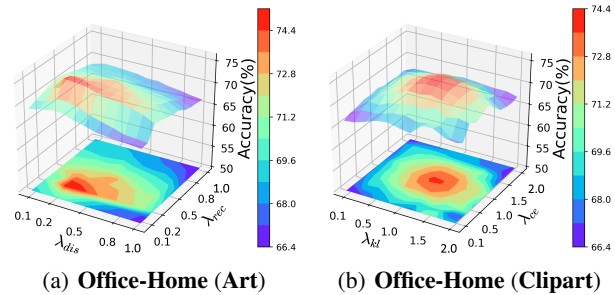

| (a) **Office-Home** (**Art**) | (b) **Office-Home** (**Clipart**) |

*Figure 4.* The sensitivity analysis ($\lambda_{dis}$, $\lambda_{rec}$, $\lambda_{ce}$, $\lambda_{kl}$) of U³CF on Office-Home dataset in terms of parameter variations.

step 1 and step 2, and step 2 depends on step 1 so we do not evaluate it in isolation.

**Effectiveness of Cross-Domain Alignment (Step 1).** As shown in Table 6, U³CF-3 outperforms U³CF-2 by 2.1% in average accuracy, indicating that removing stage 1 (initialization of $\mathcal{M}$) degrades step 1 by 2.1%. This suggests that the proposed progressive class prototype-oriented cross-domain alignment, which leverages statistical information, benefits from an accurate initialization of $\mathcal{M}$. Moreover, U³CF-3 exceeds U³CF-1 by 12.2% in average accuracy, showing that removing stage 2 (updating of $\mathcal{M}$ and cross-domain alignment) reduces step 1 performance by 12.2%. These results highlight the effectiveness of the alignment loss in step 1 in improving model accuracy.

**Effectiveness of Causal Classification (Step 2).** As reported in Table 6, U³CF-3 achieves an average accuracy of 86.4%. The proposed U³CF further improves upon U³CF-3 by 1.7%, confirming that the domain-unbiased causal classification in step 2 provides additional gains. Specifically, by disentangling features and constructing a causal classifier via the SCM, step 2 yields consistent accuracy improvements for MTDA.

### 3.5. Analysis of Parameter Sensitivity

In this part, we examine how the hyperparameters of U³CF affect performance across tasks. We vary $\lambda_{dis}$, $\lambda_{rec}$, $\lambda_{ce}$, and $\lambda_{kl}$ over $\lambda_{dis} \in \{0.1, 0.2, 0.5, 0.8, 1.0\}$, $\lambda_{rec} \in \{0.1, 0.2, 0.5, 0.8, 1.0\}$, $\lambda_{ce} \in \{0.1, 0.5, 1.0, 1.5, 2.0\}$, and $\lambda_{kl} \in \{0.1, 0.5, 1.0, 1.5, 2.0\}$. As shown in Fig. 4(a), in the task Art (Office-Home), U³CF achieves satisfactory accuracy when $\lambda_{dis} \in [0.2, 0.4]$ and $\lambda_{rec} \in [0.2, 0.3]$. As shown in Fig. 4(b), in the task Clipart (Office-Home), the model U³CF achieves remarkable accuracy when $\lambda_{kl} \in [0.9, 1.3]$ and $\lambda_{ce} \in [0.8, 1.2]$.

In our experiments, the remaining hyperparameters exhibit relatively stable behavior within reasonable ranges, and we therefore omit their sensitivity curves for brevity. The specific hyperparameter settings are provided in Appendix D.

*Table 7.* Grad-CAM visualization on **Office-31**.

| Tasks | Office-31 (Amazon) | | | Office-31 (Webcam) | | |
|---|---|---|---|---|---|---|
| Categories | **bike** | **lamp** | **chair** | **bike** | **lamp** | **chair** |
| Original Images | | | | | | |
| CDAN (Long et al., 2018) | | | | | | |
| U$^3$CF (Ours) | | | | | | |

## 3.6. Visualization Analysis

As shown in Table 7, Grad-CAM (Selvaraju et al., 2020) is utilized to visualize heatmaps of target samples for the MTDA tasks Amazon and Webcam in the Office-31 dataset. For the bike, lamp, and chair categories, the visualization shows that U$^3$CF effectively captures category-related regions and discriminative object parts. Compared with CDAN, U$^3$CF reduces attention to irrelevant background regions and focuses more consistently on semantic regions. These results indicate that U$^3$CF learns more reliable and transferable representations, which helps improve classification performance across target domains.

Additional visualizations are included in Appendix D.

## 4. Conclusion

We proposed **U$^3$CF**: Unbiased, Unconfounding, and Unified Causal Framework for MTDA. **Unbiased** prediction is derived from a causal formulation: by performing a conditional intervention and causally marginalizing the domain confounder via averaging over domain prototypes to reduce domain preference. **Unconfounding** is realized by identifying domain-specific factors as confounders in an MTDA-tailored SCM and separating them from invariant features through mutual-information-guided disentanglement. **Unified** refers to integrating progressive prototype alignment, disentanglement, and causal classification into a single end-to-end framework. Experiments show consistent improvements over strong baselines on MTDA benchmarks. We hope U$^3$CF offers a practical and principled recipe for causality-driven adaptation under multiple target shifts.

## Acknowledgements

This work was supported by the Beijing Smart Agriculture Innovation Consortium Project (No. BAIC10-2026-E14). Wenxu Wang was also supported by the Natural Science Foundation of Shandong Province (No. ZR2025QC699) and the China Postdoctoral Science Foundation (No. 2025M771557).

## Impact Statement

This paper presents work whose goal is to advance the field of Machine Learning. There are many potential societal consequences of our work, none of which we feel must be specifically highlighted here.

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

# ————Appendix————

The structure of the Appendix is as follows:

- Appendix A provides all complete *Proofs of the Propositions* in the main manuscript.

- Appendix B provides the *Algorithm Pseudocode*.

- Appendix C provides the *Related Work*.

- Appendix D provides *Additional Experiments and Implementation Details*, including dataset descriptions, implementation settings, and supplementary results.

## A. Proofs of the Propositions

### A.1. Proof of Proposition 1

**Setup.**  For a given target feature $\mathbf{z}_{ti}$, we consider $K$ prototypes $\{\mathbf{c}_s^k\}_{k=1}^K$, where $\mathbf{c}_s^{k+}$ forms the positive pair with $\mathbf{z}_{ti}$. We score each candidate prototype using $f(\mathbf{z}_{ti}, \mathbf{c}_s^k) = \exp(\mathbf{z}_{ti}^\top \mathbf{c}_s^k)$ and treat the positive index as uniformly distributed over the $K$ candidates. To obtain a tractable probabilistic interpretation, we view the remaining $K - 1$ prototypes appearing in the denominator as negative candidates generated independently of $\mathbf{z}_{ti}$ by a prototype sampling distribution (e.g., the marginal over prototypes). Under this generative view, a positive pair $(\mathbf{z}_{ti}, \mathbf{c}_s^{k+})$ reflects their same-category dependence, whereas negative pairs $(\mathbf{z}_{ti}, \mathbf{c}_s^k)$ are modeled as independent draws, which leads to the factorization used in the subsequent Bayes derivation.

**Step1: From Eq. (1) to a probabilistic form.**  Let $\mathcal{L}_{ali}(\mathbf{z}_{ti})$ be the per-sample loss term in Eq. (1):

$$\mathcal{L}_{ali}(\mathbf{z}_{ti}) = -\log \frac{f\left(\mathbf{z}_{ti}, \mathbf{c}_s^{k+}\right)}{\sum_{k=1}^K f\left(\mathbf{z}_{ti}, \mathbf{c}_s^k\right)}. \tag{22}$$

Then the overall loss is $\mathcal{L}_{ali} = \sum_{i=1}^{n_t} \mathcal{L}_{ali}(\mathbf{z}_{ti})$. Eq. (1) can be equivalently viewed as a $K$-way contrastive classification objective by defining the induced posterior

$$\mathcal{L}_{ali} = -\sum_{i=1}^{n_t} \log\left[p\left(k = y_{ti} \mid \mathbf{z}_{ti}, \mathbf{c}_s^k\right)\right], \tag{23}$$

where the uniform prior implies $p(k = y_{ti}) = \frac{1}{K}$ and $p(k \neq y_{ti}) = \frac{K-1}{K}$.

**Step2: Bayes derivation.**  Applying Bayes' rule yields

$$
\begin{aligned}
&p\left(k = y_{ti} \mid \mathbf{z}_{ti}, \mathbf{c}_s^k\right) \\
&= \frac{p\left(\mathbf{z}_{ti}, \mathbf{c}_s^k \mid k = y_{ti}\right) p\left(k = y_{ti}\right)}{p\left(\mathbf{z}_{ti}, \mathbf{c}_s^k \mid k = y_{ti}\right) p\left(k = y_{ti}\right) + p\left(\mathbf{z}_{ti}, \mathbf{c}_s^k \mid k \neq y_{ti}\right) p\left(k \neq y_{ti}\right)} \\
&= \frac{p\left(\mathbf{z}_{ti}, \mathbf{c}_s^k \mid k = y_{ti}\right)}{p\left(\mathbf{z}_{ti}, \mathbf{c}_s^k \mid k = y_{ti}\right) + p\left(\mathbf{z}_{ti}, \mathbf{c}_s^k \mid k \neq y_{ti}\right)(K - 1)}.
\end{aligned}
\tag{24}
$$

For notational convenience, we denote the joint distribution of positive pairs by $p\left(\mathbf{z}_{ti}, \mathbf{c}_s^k\right) \triangleq p\left(\mathbf{z}_{ti}, \mathbf{c}_s^k \mid k = y_{ti}\right)$. For negative pairs, we use the factorized model $p(\mathbf{z}_{ti}, \mathbf{c}_s^k \mid k \neq y_{ti}) = p(\mathbf{z}_{ti})p(\mathbf{c}_s^k)$, which matches the generative view described in the setup. Substituting these into Eq. (24) gives

$$p\left(k = y_{ti} \mid \mathbf{z}_{ti}, \mathbf{c}_s^k\right) = \frac{p\left(\mathbf{z}_{ti}, \mathbf{c}_s^k\right)}{p\left(\mathbf{z}_{ti}, \mathbf{c}_s^k\right) + p\left(\mathbf{z}_{ti}\right) p\left(\mathbf{c}_s^k\right)(K - 1)}. \tag{25}$$

**Step3: Connection to density ratios and mutual information.** In the following, we focus on the positive class for $\mathbf{z}_{ti}$ and use $k$ to denote its positive index (i.e., $\mathbf{c}_s^k$ is the positive prototype for $\mathbf{z}_{ti}$). Define the density ratio

$$r(\mathbf{z}_{ti}, \mathbf{c}_s^k) = \frac{p(\mathbf{z}_{ti}, \mathbf{c}_s^k)}{p(\mathbf{z}_{ti})p(\mathbf{c}_s^k)}. \tag{26}$$

Using Eq. (25), the posterior can be rewritten as:

$$p\left(k = y_{ti} \mid \mathbf{z}_{ti}, \mathbf{c}_s^k\right) = \frac{r(\mathbf{z}_{ti}, \mathbf{c}_s^k)}{r(\mathbf{z}_{ti}, \mathbf{c}_s^k) + (K-1)}. \tag{27}$$

Substituting Eq. (27) into Eq. (23) yields an equivalent form of the loss:

$$\begin{aligned}
\mathcal{L}_{ali} &= -\sum_{i=1}^{n_t} \log\left[\frac{r(\mathbf{z}_{ti}, \mathbf{c}_s^k)}{r(\mathbf{z}_{ti}, \mathbf{c}_s^k) + (K-1)}\right] \\
&= \sum_{i=1}^{n_t} \log\left[1 + \frac{K-1}{r(\mathbf{z}_{ti}, \mathbf{c}_s^k)}\right].
\end{aligned} \tag{28}$$

Let $\mathrm{PMI}(\mathbf{z}_{ti}, \mathbf{c}_s^k) \triangleq \log r(\mathbf{z}_{ti}, \mathbf{c}_s^k) = \log \frac{p(\mathbf{z}_{ti}, \mathbf{c}_s^k)}{p(\mathbf{z}_{ti})p(\mathbf{c}_s^k)}$ denote the Pointwise Mutual Information (PMI). Then Eq. (28) can be written as:

$$\mathcal{L}_{ali} = \sum_{i=1}^{n_t} \log\left[1 + (K-1)\exp\left(-\mathrm{PMI}(\mathbf{z}_{ti}, \mathbf{c}_s^k)\right)\right]. \tag{29}$$

Since $\log\left[1 + (K-1)\exp\left(-\mathrm{PMI}(\mathbf{z}_{ti}, \mathbf{c}_s^k)\right)\right]$ decreases as $\mathrm{PMI}(\mathbf{z}_{ti}, \mathbf{c}_s^k)$ increases, minimizing $\mathcal{L}_{ali}$ encourages larger $\mathrm{PMI}(\mathbf{z}_{ti}, \mathbf{c}_s^k)$, i.e., a larger density ratio between the joint distribution and the product of marginals.

**Step4: Mutual-information lower bound.** Under the generative view described in the setup (uniform positive index and independent negatives), the above $K$-way contrastive classification objective admits the following standard bound:

$$I\left(\mathbf{z}_{ti}; \mathbf{c}_s^{k+}\right) \geq \log K - \mathbb{E}[\mathcal{L}_{ali}(\mathbf{z}_{ti})], \tag{30}$$

which implies that minimizing $\mathcal{L}_{ali}$ increases a lower bound on the mutual information between $\mathbf{z}_{ti}$ and the same-category prototype $\mathbf{c}_s^{k+}$ (up to the additive constant $\log K$).

### A.2. Proof of Proposition 2

Set $F_i = \mathbf{f}_i$. Under the conditional intervention $\sigma_X = do(X = g(F_i))$, the structural mechanism of $X$ is replaced by the deterministic policy $g(F_i)$. Equivalently, the intervention conditional of $X$ satisfies

$$P(\mathbf{x} \mid F_i = \mathbf{f}_i, F_s = \mathbf{f}_s; \sigma_X) = \delta_{\mathbf{x}=g(\mathbf{f}_i)}. \tag{31}$$

**Step 1: Marginalize the domain-specific confounder.** By the law of total probability under the interventional regime,

$$\begin{aligned}
P(Y \mid F_i = \mathbf{f}_i; \sigma_X) &= \sum_{F_s} P(Y, F_s \mid F_i = \mathbf{f}_i; \sigma_X) \\
&= \sum_{F_s} P(Y \mid F_s, F_i = \mathbf{f}_i; \sigma_X) P(F_s \mid F_i = \mathbf{f}_i; \sigma_X).
\end{aligned} \tag{32}$$

**Step 2: Reduce $P(F_s \mid F_i; \sigma_X)$ to $P(F_s)$.** Since $\sigma_X$ only modifies the mechanism of $X$, variables other than $X$ keep their original mechanisms. In particular, $F_s$ is not intervened, hence the conditional distribution of $F_s$ given $F_i$ is unchanged:

$$P(F_s \mid F_i = \mathbf{f}_i; \sigma_X) = P(F_s \mid F_i = \mathbf{f}_i). \tag{33}$$

Moreover, in the SCM of Fig. (b)–(c) we model $F_s$ and $F_i$ as independent parent factors, so $P(F_s \mid F_i = \mathbf{f}_i) = P(F_s)$. Substituting into Eq. (32) yields

$$P(Y \mid F_i = \mathbf{f}_i; \sigma_X) = \sum_{F_s} P(Y \mid F_s, F_i = \mathbf{f}_i; \sigma_X) P(F_s). \tag{34}$$

**Step 3: Plug in the deterministic policy for $X$.** We expand $P(Y \mid F_s, F_i = \mathbf{f}_i; \sigma_X)$ by marginalizing $X$:

$$
\begin{aligned}
P(Y \mid F_s, F_i = \mathbf{f}_i; \sigma_X) &= \sum_{\mathbf{x}} P(Y \mid X = \mathbf{x}, F_s, F_i = \mathbf{f}_i; \sigma_X)\, P(\mathbf{x} \mid F_s, F_i = \mathbf{f}_i; \sigma_X) \\
&= \sum_{\mathbf{x}} P(Y \mid X = \mathbf{x}, F_s, F_i = \mathbf{f}_i)\, \delta_{\mathbf{x}=g(\mathbf{f}_i)} \\
&= P(Y \mid F_s, F_i = \mathbf{f}_i, X = g(\mathbf{f}_i)),
\end{aligned}
\tag{35}
$$

where the second line uses Eq. (31) and the fact that $\sigma_X$ does not modify the mechanism of $Y$.

Combining Eq. (34) and Eq. (35) gives

$$
P(Y \mid F_i = \mathbf{f}_i; \sigma_X) = \sum_{F_s} P(Y \mid F_s, F_i = \mathbf{f}_i, X = g(\mathbf{f}_i))\, P(F_s),
\tag{36}
$$

which proves the first claim in Proposition 2.

**Step 4: Domain-prototype approximation (uniform averaging).** If $P(F_s)$ is approximated by $K_D$ domain prototypes $\{\widehat{\mathbf{f}}_{sj}\}_{j=1}^{K_D}$ with uniform weights $P(F_s = \widehat{\mathbf{f}}_{sj}) = \frac{1}{K_D}$, then

$$
\begin{aligned}
P(Y \mid F_i = \mathbf{f}_i; \sigma_X) &= \sum_{j=1}^{K_D} P\Big(Y \mid F_s = \widehat{\mathbf{f}}_{sj}, F_i = \mathbf{f}_i, X = g(\mathbf{f}_i)\Big)\, P(F_s = \widehat{\mathbf{f}}_{sj}) \\
&= \frac{1}{K_D} \sum_{j=1}^{K_D} P\Big(Y \mid F_s = \widehat{\mathbf{f}}_{sj}, F_i = \mathbf{f}_i, X = g(\mathbf{f}_i)\Big),
\end{aligned}
\tag{37}
$$

which proves Eq. (4) and completes the proof.

### A.3. Proof of Proposition 3

We work under the data distribution $p(\mathbf{x})$ and the encoder-induced posteriors $p(\mathbf{f}_i \mid \mathbf{x})$ and $p(\mathbf{f}_s \mid \mathbf{x})$. We adopt a factorized inference model

$$
p(\mathbf{f}_i, \mathbf{f}_s \mid \mathbf{x}) = p(\mathbf{f}_i \mid \mathbf{x})\, p(\mathbf{f}_s \mid \mathbf{x}),
\tag{38}
$$

which implies $I(\mathbf{f}_i; \mathbf{f}_s \mid \mathbf{x}) = 0$.

**Step 1: Mutual-information decomposition via interaction information.** We aim to minimize $I(\mathbf{f}_i; \mathbf{f}_s)$ so that $\mathbf{f}_i$ and $\mathbf{f}_s$ capture complementary factors. Recall the interaction information identity among three random variables $(X, Y, Z)$ (McGill, 1954; Hwang et al., 2020):

$$
\begin{aligned}
I(X; Y; Z) &= I(X; Y) - I(X; Y \mid Z) \\
&= I(X; Z) - I(X; Z \mid Y) \\
&= I(Y; Z) - I(Y; Z \mid X).
\end{aligned}
\tag{39}
$$

Using the last equality with $(X, Y, Z) = (\mathbf{x}, \mathbf{f}_i, \mathbf{f}_s)$ gives

$$
I(\mathbf{f}_i; \mathbf{f}_s) = I(\mathbf{f}_i; \mathbf{x}) - I(\mathbf{f}_i; \mathbf{x} \mid \mathbf{f}_s) + I(\mathbf{f}_i; \mathbf{f}_s \mid \mathbf{x}).
\tag{40}
$$

By Eq. (38), the conditional mutual information vanishes:

$$
\begin{aligned}
I(\mathbf{f}_i; \mathbf{f}_s \mid \mathbf{x}) &= H(\mathbf{f}_i \mid \mathbf{x}) - H(\mathbf{f}_i \mid \mathbf{f}_s, \mathbf{x}) \\
&= H(\mathbf{f}_i \mid \mathbf{x}) - H(\mathbf{f}_i \mid \mathbf{x}) = 0.
\end{aligned}
\tag{41}
$$

Substituting Eq. (41) into Eq. (40) yields

$$
\begin{aligned}
I(\mathbf{f}_i; \mathbf{f}_s) &= I(\mathbf{x}; \mathbf{f}_i) - I(\mathbf{x}; \mathbf{f}_i \mid \mathbf{f}_s) \\
&= I(\mathbf{x}; \mathbf{f}_i) + I(\mathbf{x}; \mathbf{f}_s) - I(\mathbf{x}; \mathbf{f}_i, \mathbf{f}_s),
\end{aligned}
\tag{42}
$$

where the last equality uses the chain rule $I(\mathbf{x}; \mathbf{f}_i, \mathbf{f}_s) = I(\mathbf{x}; \mathbf{f}_s) + I(\mathbf{x}; \mathbf{f}_i \mid \mathbf{f}_s)$. This proves the decomposition part of Proposition 3.

**Step 2: Detailed variational upper bound for $I(\mathbf{x}; \mathbf{f}_i)$.** By definition,

$$I(\mathbf{x}; \mathbf{f}_i) = \mathbb{E}_{p(\mathbf{x}, \mathbf{f}_i)} \left[ \log \frac{p(\mathbf{f}_i \mid \mathbf{x})}{p(\mathbf{f}_i)} \right], \tag{43}$$

where $p(\mathbf{f}_i) = \int p(\mathbf{f}_i \mid \mathbf{x}) p(\mathbf{x}) \, d\mathbf{x}$ is the marginal. Introduce a variational marginal $q(\mathbf{f}_i)$ and multiply/divide by it:

$$
\begin{aligned}
I(\mathbf{x}; \mathbf{f}_i) &= \mathbb{E}_{p(\mathbf{x}, \mathbf{f}_i)} \left[ \log \frac{p(\mathbf{f}_i \mid \mathbf{x}) q(\mathbf{f}_i)}{q(\mathbf{f}_i) p(\mathbf{f}_i)} \right] \\
&= \mathbb{E}_{p(\mathbf{x}, \mathbf{f}_i)} \left[ \log \frac{p(\mathbf{f}_i \mid \mathbf{x})}{q(\mathbf{f}_i)} \right] - \mathbb{E}_{p(\mathbf{f}_i)} \left[ \log \frac{p(\mathbf{f}_i)}{q(\mathbf{f}_i)} \right] \\
&= \mathbb{E}_{p(\mathbf{x})} \left[ D_{\mathrm{KL}} \big( p(\mathbf{f}_i \mid \mathbf{x}) \, \| \, q(\mathbf{f}_i) \big) \right] - D_{\mathrm{KL}} \big( p(\mathbf{f}_i) \, \| \, q(\mathbf{f}_i) \big).
\end{aligned} \tag{44}
$$

Dropping the nonnegative term $D_{\mathrm{KL}}(p(\mathbf{f}_i) \| q(\mathbf{f}_i)) \geq 0$ gives

$$I(\mathbf{x}; \mathbf{f}_i) \leq \mathbb{E}_{p(\mathbf{x})} \left[ D_{\mathrm{KL}} \big( p(\mathbf{f}_i \mid \mathbf{x}) \, \| \, q(\mathbf{f}_i) \big) \right] \triangleq \mathcal{L}_{dis-\mathbf{f}_i}. \tag{45}$$

**Step 3: Detailed variational upper bound for $I(\mathbf{x}; \mathbf{f}_s)$.** Repeating the same derivation with $q(\mathbf{f}_s)$ yields

$$I(\mathbf{x}; \mathbf{f}_s) \leq \mathbb{E}_{p(\mathbf{x})} \left[ D_{\mathrm{KL}} \big( p(\mathbf{f}_s \mid \mathbf{x}) \, \| \, q(\mathbf{f}_s) \big) \right] \triangleq \mathcal{L}_{dis-\mathbf{f}_s}. \tag{46}$$

**Step 4: Detailed variational lower bound for $I(\mathbf{x}; \mathbf{f}_i, \mathbf{f}_s)$.** Let $\mathbf{z} \triangleq (\mathbf{f}_i, \mathbf{f}_s)$. Because $(\mathbf{x}, \mathbf{z})$ is induced by the encoder at training time, we define the encoder-induced joint

$$\tilde{p}(\mathbf{x}, \mathbf{z}) \triangleq p(\mathbf{x}) \, p(\mathbf{z} \mid \mathbf{x}) = p(\mathbf{x}) \, p(\mathbf{f}_i \mid \mathbf{x}) \, p(\mathbf{f}_s \mid \mathbf{x}), \tag{47}$$

and consider the mutual information under $\tilde{p}$:

$$I(\mathbf{x}; \mathbf{z}) = \mathbb{E}_{\tilde{p}(\mathbf{x}, \mathbf{z})} \left[ \log \frac{\tilde{p}(\mathbf{x} \mid \mathbf{z})}{p(\mathbf{x})} \right] = H(\mathbf{x}) + \mathbb{E}_{\tilde{p}(\mathbf{x}, \mathbf{z})} \left[ \log \tilde{p}(\mathbf{x} \mid \mathbf{z}) \right]. \tag{48}$$

Since $\tilde{p}(\mathbf{x} \mid \mathbf{z})$ is intractable, introduce a variational decoder $q(\mathbf{x} \mid \mathbf{z})$. Using the standard variational identity,

$$
\begin{aligned}
\mathbb{E}_{\tilde{p}(\mathbf{x}, \mathbf{z})} \left[ \log \tilde{p}(\mathbf{x} \mid \mathbf{z}) \right] &= \mathbb{E}_{\tilde{p}(\mathbf{x}, \mathbf{z})} \left[ \log q(\mathbf{x} \mid \mathbf{z}) \right] + \mathbb{E}_{\tilde{p}(\mathbf{z})} \left[ D_{\mathrm{KL}} \big( \tilde{p}(\mathbf{x} \mid \mathbf{z}) \, \| \, q(\mathbf{x} \mid \mathbf{z}) \big) \right] \\
&\geq \mathbb{E}_{\tilde{p}(\mathbf{x}, \mathbf{z})} \left[ \log q(\mathbf{x} \mid \mathbf{z}) \right],
\end{aligned} \tag{49}
$$

where the inequality follows from KL nonnegativity. Substituting Eq. (49) into Eq. (48) yields

$$I(\mathbf{x}; \mathbf{f}_i, \mathbf{f}_s) \geq H(\mathbf{x}) + \mathbb{E}_{p(\mathbf{x})} \left[ \mathbb{E}_{p(\mathbf{f}_i \mid \mathbf{x}) p(\mathbf{f}_s \mid \mathbf{x})} \left[ \log q(\mathbf{x} \mid \mathbf{f}_i, \mathbf{f}_s) \right] \right]. \tag{50}$$

To match minimization, define the reconstruction loss as the negative expected log-likelihood:

$$\mathcal{L}_{rec} \triangleq - \mathbb{E}_{p(\mathbf{x})} \left[ \mathbb{E}_{p(\mathbf{f}_i \mid \mathbf{x}) p(\mathbf{f}_s \mid \mathbf{x})} \left[ \log q(\mathbf{x} \mid \mathbf{f}_i, \mathbf{f}_s) \right] \right]. \tag{51}$$

Then Eq. (50) becomes

$$I(\mathbf{x}; \mathbf{f}_i, \mathbf{f}_s) \geq H(\mathbf{x}) - \mathcal{L}_{rec}. \tag{52}$$

*Remark.* The term $-\mathcal{L}_{rec}$ is exactly the expected reconstruction log-likelihood used in VAE-style ELBO objectives; here it appears as a variational lower bound on the joint mutual information.

**Step 5: Combine bounds to upper-bound $I(\mathbf{f}_i; \mathbf{f}_s)$.** Plugging Eqs. (45), (46), and (52) into Eq. (42), we obtain

$$
\begin{aligned}
I(\mathbf{f}_i; \mathbf{f}_s) &= I(\mathbf{x}; \mathbf{f}_i) + I(\mathbf{x}; \mathbf{f}_s) - I(\mathbf{x}; \mathbf{f}_i, \mathbf{f}_s) \\
&\leq \mathcal{L}_{dis-\mathbf{f}_i} + \mathcal{L}_{dis-\mathbf{f}_s} + \mathcal{L}_{rec} - H(\mathbf{x}) = \mathcal{L}_{dis} - H(\mathbf{x}),
\end{aligned} \tag{53}
$$

where $\mathcal{L}_{dis}$ is defined in Eq. (10) as $\mathcal{L}_{dis} \triangleq \mathcal{L}_{dis-\mathbf{f}_i} + \mathcal{L}_{dis-\mathbf{f}_s} + \mathcal{L}_{rec}$. Since $H(\mathbf{x})$ is constant w.r.t. the model parameters, minimizing $\mathcal{L}_{dis}$ minimizes an upper bound on $I(\mathbf{f}_i; \mathbf{f}_s)$. This completes the proof.

## A.4. Proof of Proposition 4

We work under the joint distribution $p(\mathbf{f}, \mathbf{y})$ induced by the data and the current encoder, where $\mathbf{y}$ is a discrete label taking values in $\mathcal{Y}$. Recall the mutual information identities

$$I(\mathbf{f}; \mathbf{y}) = \mathbb{E}_{p(\mathbf{f}, \mathbf{y})}\left[\log \frac{p(\mathbf{y} \mid \mathbf{f})}{p(\mathbf{y})}\right] = H(\mathbf{y}) + \mathbb{E}_{p(\mathbf{f}, \mathbf{y})}\left[\log p(\mathbf{y} \mid \mathbf{f})\right], \tag{54}$$

$$I(\mathbf{f}; \mathbf{y}) = \mathbb{E}_{p(\mathbf{f})}\left[D_{\mathrm{KL}}\big(p(\mathbf{y} \mid \mathbf{f}) \,\|\, p(\mathbf{y})\big)\right]. \tag{55}$$

**(i) Label relevance: a variational lower bound.** The conditional $p(\mathbf{y} \mid \mathbf{f})$ can be intractable in deep models. Introduce any variational classifier $q(\mathbf{y} \mid \mathbf{f})$ and use the KL nonnegativity:

$$\mathbb{E}_{p(\mathbf{f}, \mathbf{y})}\left[\log p(\mathbf{y} \mid \mathbf{f})\right] = \mathbb{E}_{p(\mathbf{f}, \mathbf{y})}\left[\log q(\mathbf{y} \mid \mathbf{f})\right] + \mathbb{E}_{p(\mathbf{f})}\left[D_{\mathrm{KL}}\big(p(\mathbf{y} \mid \mathbf{f}) \,\|\, q(\mathbf{y} \mid \mathbf{f})\big)\right]$$
$$\geq \mathbb{E}_{p(\mathbf{f}, \mathbf{y})}\left[\log q(\mathbf{y} \mid \mathbf{f})\right]. \tag{56}$$

Plugging Eq. (56) into Eq. (54) yields

$$I(\mathbf{f}; \mathbf{y}) \geq H(\mathbf{y}) + \mathbb{E}_{p(\mathbf{f}, \mathbf{y})}\left[\log q(\mathbf{y} \mid \mathbf{f})\right]. \tag{57}$$

Define the cross-entropy (negative log-likelihood) w.r.t. $q$:

$$\mathcal{L}_{\text{cross-entropy}} \triangleq -\mathbb{E}_{p(\mathbf{f}, \mathbf{y})}\left[\log q(\mathbf{y} \mid \mathbf{f})\right]. \tag{58}$$

Then Eq. (57) becomes exactly Eq. (13) in Proposition 4:

$$I(\mathbf{f}; \mathbf{y}) \geq H(\mathbf{y}) - \mathcal{L}_{\text{cross-entropy}}. \tag{59}$$

Since $H(\mathbf{y})$ depends only on the label distribution (not on model parameters), minimizing $\mathcal{L}_{\text{cross-entropy}}$ increases this lower bound and thus encourages larger $I(\mathbf{f}; \mathbf{y})$.

**(ii) Label irrelevance: an upper bound via KL-to-reference.** For any reference marginal $r(\mathbf{y})$, we start from the MI identity

$$I(\mathbf{f}; \mathbf{y}) = \mathbb{E}_{p(\mathbf{f}, \mathbf{y})}\left[\log \frac{p(\mathbf{y} \mid \mathbf{f})}{p(\mathbf{y})}\right] = \mathbb{E}_{p(\mathbf{f}, \mathbf{y})}\left[\log \frac{p(\mathbf{y} \mid \mathbf{f})}{r(\mathbf{y})}\right] - \mathbb{E}_{p(\mathbf{y})}\left[\log \frac{p(\mathbf{y})}{r(\mathbf{y})}\right]$$
$$= \mathbb{E}_{p(\mathbf{f})}\left[D_{\mathrm{KL}}\big(p(\mathbf{y} \mid \mathbf{f}) \,\|\, r(\mathbf{y})\big)\right] - D_{\mathrm{KL}}\big(p(\mathbf{y}) \,\|\, r(\mathbf{y})\big), \tag{60}$$

which is the decomposition in Eq. (14). By nonnegativity of KL, $D_{\mathrm{KL}}(p(\mathbf{y})\|r(\mathbf{y})) \geq 0$, we have the upper bound

$$I(\mathbf{f}; \mathbf{y}) \leq \mathbb{E}_{p(\mathbf{f})}\left[D_{\mathrm{KL}}\big(p(\mathbf{y} \mid \mathbf{f}) \,\|\, r(\mathbf{y})\big)\right]. \tag{61}$$

**(iii) Specializing to the uniform reference.** If we choose the uniform reference $r(\mathbf{y}) = u(\mathbf{y}) = \frac{1}{|\mathcal{Y}|}$, then Eq. (61) becomes the KL-to-uniform regularizer used in the main text. In particular, applying (i) and (ii) to the four MI objectives in Eq. (12) yields:

$$\mathcal{L}_{ce-\mathbf{f}_i} \triangleq -\mathbb{E}_{p(\mathbf{f}_i, \mathbf{y}_c)}\left[\log q(\mathbf{y}_c \mid \mathbf{f}_i)\right], \qquad \mathcal{L}_{ce-\mathbf{f}_s} \triangleq -\mathbb{E}_{p(\mathbf{f}_s, \mathbf{y}_d)}\left[\log q(\mathbf{y}_d \mid \mathbf{f}_s)\right], \tag{62}$$

$$\mathcal{L}_{kl-\mathbf{f}_i} \triangleq \mathbb{E}_{p(\mathbf{f}_i)}\left[D_{\mathrm{KL}}\big(p(\mathbf{y}_d \mid \mathbf{f}_i) \,\|\, u_d(\mathbf{y}_d)\big)\right], \qquad \mathcal{L}_{kl-\mathbf{f}_s} \triangleq \mathbb{E}_{p(\mathbf{f}_s)}\left[D_{\mathrm{KL}}\big(p(\mathbf{y}_c \mid \mathbf{f}_s) \,\|\, u_c(\mathbf{y}_c)\big)\right], \tag{63}$$

where $u_d(\mathbf{y}_d) = \frac{1}{K_D}$ and $u_c(\mathbf{y}_c) = \frac{1}{K}$. Therefore, the four tractable losses in Eqs. (15)–(18) are exactly the variational surrogates implied by Proposition 4.

# B. Algorithm Pseudocode

The overall procedure is summarized in Algorithm 1.

---

**Algorithm 1:** The proposed U$^3$CF

---

/* **Step 1-Stage 1: Initialize** $\mathcal{M}$ */
**Input:** source domain $\mathcal{D}_S$, target domains $\mathcal{D}_T$; domain number $K_D$, category number $K$; domain labels $\mathcal{Y}_d$, category labels $\mathcal{Y}_c$; parameters $\lambda_{node}$.
**while** *the maximum number of iterations is not reached* **do**
    Initialize $\mathcal{M} \leftarrow$ pre-trained $G_\theta$.
    Train $G_\theta, C_{mlp}, C_{gcn} \leftarrow \mathcal{L}_{train} = \mathcal{L}_{ce}^{mlp} + \mathcal{L}_{bce}^{edge} + \lambda_{node}\mathcal{L}_{ce}^{node}$.
    Update $\mathcal{M} \leftarrow$ trained $G_\theta$.
**end**
**Output:** Memory module $\mathcal{M}$; Networks $G_\theta$, MLP classifier $C_{mlp}$, GCN classifier $C_{gcn}$.
/* **Step 1-Stage 2: Cross-domain alignment** */
**Input:** source domain $\mathcal{D}_S$, target domains $\mathcal{D}_T$; domain number $K_D$, category number $K$; domain labels $\mathcal{Y}_d$, category labels $\mathcal{Y}_c$; parameters $\lambda_{node}$,
    $\lambda_{ali}$, $\theta$; networks $G_\theta, C_{mlp}, C_{gcn}$; Memory module $\mathcal{M}$.
Calculate class prototype $\mathbf{c}_s^k$.
**for** $\mathcal{D}_{ti}$ in $\mathcal{D}_T$ **do**
    Train $G_\theta, C_{mlp}, C_{gcn} \leftarrow \mathcal{L}_{train} = \mathcal{L}_{ce}^{mlp} + \mathcal{L}_{bce}^{edge} + \lambda_{node}\mathcal{L}_{ce}^{node} + \lambda_{ali}\mathcal{L}_{ali}(Eq.(1))$ .
    Update $\mathcal{M} \leftarrow$ trained $G_\theta$.
    Update $\mathbf{c}_s^k \leftarrow \mathcal{M}$.
    Select $\widehat{\mathcal{D}}_{ti} \leftarrow \theta$.
    High-confidence domain $\mathcal{D}_H \leftarrow \mathcal{D}_S + \widehat{\mathcal{D}}_T$.
**end**
**Output:** Memory module $\mathcal{M}$; Networks $G_\theta$, MLP classifier $C_{mlp}$, GCN classifier $C_{gcn}$; High-confidence domain $\mathcal{D}_H$.
/* **Step 2: Causal classification** */
**Input:** source domain $\mathcal{D}_S$, target domains $\mathcal{D}_T$; domain number $K_D$, category number $K$; domain labels $\mathcal{Y}_d$, category labels $\mathcal{Y}_c$; parameters $\lambda_{node}$,
    $\lambda_{dis}, \lambda_{rec}, \lambda_{ce}, \lambda_{kl}$; networks $G_\theta, C_{mlp}, C_{gcn}$; Memory module $\mathcal{M}$; High-confidence domain $\mathcal{D}_H$.
Calculate confounder $\widehat{\mathbf{f}}_{sd}$.
**while** *the maximum number of iterations is not reached* **do**
    Train $G_\theta, F_i, F_s, C_c, C_d, G_r, C_{mlp}, C_{gcn} \leftarrow \mathcal{L}_{train} = \mathcal{L}_{cls-m}^{causal}(Eq.(19)) + \mathcal{L}_{bce}^{edge} + \lambda_{node}\mathcal{L}_{cls-g}^{causal}(Eq.(20)) + \mathcal{L}_{dis}(Eq.(10)) +$
    $\lambda_{ce}(\mathcal{L}_{ce-\mathbf{f}_i}(Eq.(15)) + \mathcal{L}_{ce-\mathbf{f}_s}(Eq.(16))) + \lambda_{kl}(\mathcal{L}_{kl-\mathbf{f}_i}(Eq.(17)) + \mathcal{L}_{kl-\mathbf{f}_s}(Eq.(18)))$.
    Update $\mathcal{M} \leftarrow$ domain-specific features $\mathbf{f}_s$.
    Update $\widehat{\mathbf{f}}_{sd} \leftarrow \mathcal{M}$.
**end**
**Output:** MLP classifier $C_{mlp}$, GCN classifier $C_{gcn}$.
/* **Testing phase.** */
$\mathbf{y}_{gcn}^{causal} \leftarrow$ GCN classifier $C_{gcn}$ (Eq.(21))
Return: $\mathbf{y}_{gcn}^{causal}$

---

# C. Related Work

## C.1. Single-Target Domain Adaptation (STDA)

The STDA paradigm has been extensively studied, and the main methods include statistical information (Chen et al., 2020; Wang et al., 2023a; 2024; Long et al., 2019; 2017; Pan et al., 2011; Long et al., 2013), adversarial learning (Ganin & Lempitsky, 2015; Long et al., 2018; Bousmalis et al., 2017; Kang et al., 2018; Murez et al., 2018), and generative model (Liu et al., 2019; Wang et al., 2019; Chang et al., 2019; Li et al., 2021; Wang et al., 2023b), and others. In methods based on statistical information, Pan et al. (Pan et al., 2011) and Long et al. (Long et al., 2013) first proposed to align the marginal and conditional distributions. Further, Long et al. (Long et al., 2019; 2017) integrated metric learning with neural networks. Chen et al. (Chen et al., 2020) incorporated metric learning into graph embedding structures. Wang et al. (Wang et al., 2023a; 2024) achieved homogenous and heterogeneous domain adaptation through graph embedding structures. In adversarial learning methods, Ganin et al. (Ganin & Lempitsky, 2015) and Long et al. (Long et al., 2018) utilized the gradient reversal layer (GRL) to learn domain-invariant features. Some approaches achieve domain adaptation based on GAN networks. Bousmalis et al. (Bousmalis et al., 2017) implemented cross-domain target style transfer. Kang et al. (Kang et al., 2018) achieved cross-domain consistency through a source-to-target or target-to-source cycle. Murez et al. (Murez et al., 2018) proposed a unified framework integrating domain-agnostic feature extraction with domain generation. In generative model-based methods, Li et al. (Li et al., 2021) proposed cross-domain alignment based on implicit feature augmentation. Further, Wang et al. (Wang et al., 2023b) combined contrastive learning and adversarial learning to optimize domain adaptation methods based on implicit augmentation. Liu et al. (Liu et al., 2019) utilized orthogonal regularization to learn private and

shared features for different domains. Wang et al. (Wang et al., 2019) integrated attention mechanisms with DA tasks to make the model focus on transferable regions in images. Chang et al. (Chang et al., 2019) utilized batch normalization for different domains to learn domain-invariant features. To explore feature disentanglement mechanisms, Deng et al. (Deng et al., 2021; 2022) employed variational information bottleneck theory (VIB) (Alemi et al., 2017) to learn domain-invariant and domain-specific features. To explore causality in domain adaptation, some researchers have introduced causal inference into the domain adaptation (DA) tasks. Yue et al. (Yue et al., 2021) constructed a structural causal model for STDA tasks to achieve domain adaptation. Inspired by causality, Lv et al. (Lv et al., 2022) learned the representations suitable for OOD data. Mao et al. (Mao et al., 2022) and Mahajan et al. (Mahajan et al., 2021) further explored SCMs applicable to domain generalization. Li et al. (Li et al., 2026) presented a unified survey of CLIP-powered domain generalization and domain adaptation, proposing a comprehensive scenario taxonomy and summarizing representative methods, trends, challenges, and future directions for improving domain robustness. Vuong et al. (Vuong et al., 2025) improved CLIP-based UDA by reinforcing pseudo-labels and target-prompt learning via geometry-aware alignment between visual and text embeddings, including an optimal-transport strategy that enforces clustering to better match target-domain representations.

### C.2. Multi-Target Domain Adaptation (MTDA)

Due to its increasing applicability, MTDA is receiving increasing attention from researchers, making it a worthy direction for study. Huang et al. (Huang et al., 2024) weighted different target domains and proposed a traditional MTDA method based on evidence theory. Gholami et al. (Gholami et al., 2020) proposed an information theoretic method to find a shared latent space common to all domains. Nguyen-Meidine et al. (Nguyen-Meidine et al., 2021) utilized a multi-teacher knowledge distillation strategy, alternating between teachers to preserve each target's uniqueness during the student's adaptation. Chen et al. (Chen et al., 2019) implemented two adversarial processes: the first to bridge the source and mixed target domains, and the second deploying an unsupervised meta-learner that receives target data along with continuous feedback on feature learning. Roy et al. (Roy et al., 2021) proposed a graph convolutional network (GCN) as a classifier for MTDA, combined with the CDAN (Long et al., 2018) method for adversarial domain adaptation. Xu et al. (Xu et al., 2023) focused on blended target domain adaptation, aligning both low-level and high-level features, and proposed the view that category alignment is important. Lu et al. (Lu et al., 2024) proposed a dynamic feature generator with an attention mechanism for multi-source multi-target domain adaptation, enabling adaptive extraction of domain-agnostic features and improved cross-domain generalization. Wei et al. (Wei et al., 2024) proposed an incremental unsupervised MTDA framework that uses contrastive adaptation per target while preserving past knowledge via a frozen container network to mitigate catastrophic forgetting. Lu et al. (Lu et al., 2025a) performed pair-wise source–target alignment and multi-classifier alignment, augmented with semantic discrepancy minimization and target-style transfer, to tackle multi-source multi-target domain adaptation. Shin et al. (Shin et al., 2025) introduced a merge-friendly post-training quantization method for MTDA by regularizing quantization with Hessian- and distance-aware constraints to improve model merging under low-bit settings.

Inspired by D-CGCT (Roy et al., 2021), the proposed method $U^3CF$ adopts a GCN classifier. However, unlike D-CGCT, $U^3CF$ identifies the importance of effectively leveraging domain-specific features to enhance model precision in MTDA tasks. Consequently, we propose a domain-unbiased causal classifier based on feature disentanglement. Furthermore, to mitigate the challenges posed by multiple domain shifts in MTDA tasks, we propose a novel cross-domain alignment loss.

## D. Additional Experiments and Implementation Details

### D.1. Experimental Datasets

**Office-Home** (Venkateswara et al., 2017): It is a challenging DA benchmark, consisting of 15,585 images in 65 categories across four domains: Artistic, Clip, Product, and Real-World. Four MTDA tasks are constructed by using each domain as the source and the remaining as multiple target domains.

**MTRS**[1] (Zheng et al., 2022): It is a challenging DA remote sensing scene dataset, derived from five open-source datasets: UC Merced (U), AID (A), NWPU-RESISC45 (N), RSD46-WHU (R), and PatternNet (P), consisting of more than 59,115 images across 10 object categories.

**Office-31** (Saenko et al., 2010): It is a widely-used DA benchmark, consisting of 4,110 images across 31 categories, divided into three domains: Amazon with online-sourced images, DSLR featuring high-resolution DSLR camera photos, and

---

[1]The datasets are sourced from https://github.com/rs-dl/TSAN

Webcam comprising low-resolution web camera images.

**DomainNet** (Peng et al., 2019a): It is one of the most challenging and large-scale DA benchmarks, consisting of six distinct domains: Clipart (Cli.), Infograph (Inf.), Painting (Pai.), Quickdraw (Qui.), Real (Rea.), and Sketch (Ske.). The dataset contains approximately 600,000 images, and covers 345 different object categories.

### D.2. Experimental Setup

**Comparison methods.** To evaluate the performance of the proposed U$^3$CF, we compare it with advanced DA approaches. Comparison methods include Single-Target methods, Target-Combined methods, and Multi-Target methods. Single-Target methods refer to approaches applied in a one-source-to-one-target setting, including JAN (2017) (Long et al., 2017), CDAN (2018) (Long et al., 2018), TADA (2019) (Wang et al., 2019), CGCT (2021) (Roy et al., 2021), MEDM (2023) (Wu et al., 2023), and ODJID (2024) (Ye et al., 2024). Target-Combined methods refer to approaches used in a one-source-to-aggregated-targets setting, including CDAN (2018) (Long et al., 2018), SE (2018) (French et al., 2018), MCD (2018) (Saito et al., 2018), DADA (2019) (Peng et al., 2019b), AMEAN (2019) (Chen et al., 2019), TADA (2019) (Wang et al., 2019), MCC (2020) (Jin et al., 2020), CGCT (2021) (Roy et al., 2021), DML (2022) (Wang et al., 2022), MCDA (2023) (Xu et al., 2023). Multi-Target methods refer to approaches designed for a one-source-to-multiple-targets setting, including MT-MTDA (2021) (Nguyen-Meidine et al., 2021), DCL (2021) (Roy et al., 2021), D-CGCT (2021) (Roy et al., 2021), DML (2022) (Wang et al., 2022), CoNMix (2023) (Kumar et al., 2023), CAIC (2024) (Wei et al., 2024), MAN (2025) (Lu et al., 2025a), IPCA (2025) (Lu et al., 2025b).

**Parameter Settings.** In Step 1-Stage 1: the parameters $\lambda_{node} = 0.3$; in Step 1-Stage 2: the parameters $\lambda_{node} = 0.3$, $\lambda_{ali} = 0.1$; the target sample selection threshold is $\theta = 0.8$; in Step 2: the parameters $\lambda_{node} = 0.3$, $\lambda_{dis} = 0.25$, $\lambda_{rec} = 0.2$, $\lambda_{ce} = 1$, $\lambda_{kl} = 1$. For the DomainNet, we set $\lambda_{node} = 0.5$, $\lambda_{ce} = 0.5$, $\theta = 0.85$, while keeping the other hyperparameters unchanged. Our training approach employs stochastic gradient descent (SGD) for model training. We use a learning rate of 0.001 for the feature extractor and 0.01 for the classifiers and the Feature Disentanglement Module (FDM). For both the classifiers and the FDM module, we set a momentum of 0.9, a weight decay of 0.001, a bottleneck dimension of 256, and a batch size of 32. Our experiments were conducted on the A100 (40G) and L20 (48G) GPUs using PyTorch 1.10.2 and Python 3.7.11.

**Evaluation metric.** The final experimental result is taken as the average result across five random seeds. The evaluation metric proposed is the mean classification accuracy across each target domain's data, which is defined as

$$\text{Accuracy } = \frac{1}{T} \sum_{i=1}^{T} \frac{|\mathbf{x} : \mathbf{x} \in \mathcal{D}_{ti} \wedge \widehat{y}(\mathbf{x}) = y(\mathbf{x})|}{|\mathbf{x} : \mathbf{x} \in \mathcal{D}_{ti}|} \tag{64}$$

where $\mathcal{D}_{ti}$ is the target domain, $y(\mathbf{x})$ is the ground-truth label of x, $\widehat{y}(\mathbf{x})$ is the predicted label, and $T$ is the number of the target domains.

### D.3. Visualization Analysis

To illustrate feature distributions within the latent space, T-SNE visualizations of U$^3$CF are presented in Fig. 5. Specifically, we visualize the disentangled domain-invariant features $\mathbf{f}_i$, domain-specific features $\mathbf{f}_s$, and the mean of $T$ causal features obtained through the feature reconstruction module $G_r$, denoted as $\mathbf{f}_{causal}$. Subfigures (a), (b), (c), (d) are experiments conducted on task DSLR, and subfigures (e), (f), (g), (h) are experiments conducted on task Clipart. As shown in Fig. 5(c) and 5(g), domain-specific features $\mathbf{f}_s$ are visualized, with the $\mathbf{f}_s$ categorized according to domain labels, facilitating the learning of domain prototypes. From the feature distributions depicted in Fig. 5 (a), (b), (d), (e), (f), (h) for CDAN, U$^3$CF-$\mathbf{f}_i$, and U$^3$CF-$\mathbf{f}_{causal}$, it is apparent that U$^3$CF-$\mathbf{f}_{causal}$ effectively organizes samples of the same categories across different domains. Further, it enhances the density of samples within the same category and increases the separation between samples of different categories.

### D.4. Overhead of U$^3$CF

As shown in Table 8, we provide a detailed step-wise overhead analysis of U$^3$CF. Specifically:

① **Memory footprint and parameter count.** Despite introducing the memory bank and dual classifiers, U$^3$CF does not increase peak memory usage. Its peak GPU memory is consistently lower than that of D-CGCT across all stages. This is

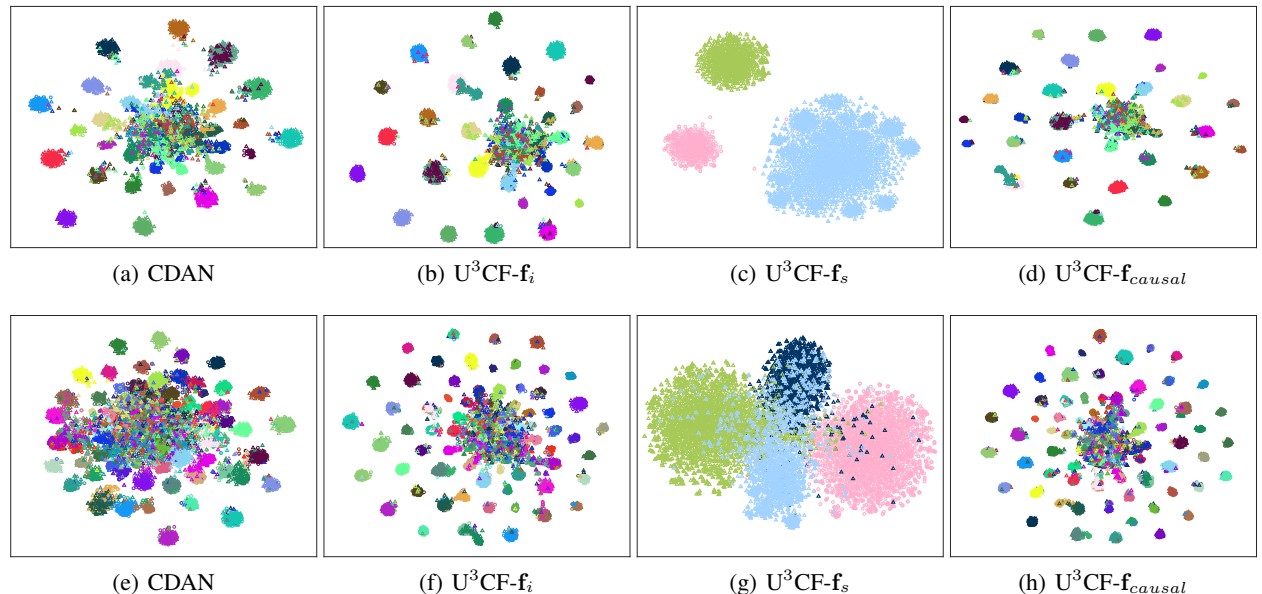

*Figure 5.* T-SNE visualizations of U$^3$CF and other methods of MTDA tasks DSLR (Office-31) and Clipart (Office-Home). In subfigures (a), (b), (d), (e), (f), (h), different colors denote different categories. In subfigures (c), (g), different colors denote different domains. The markers "o" and "△" represent the source and target domains respectively.

*Table 8.* Step-wise overhead analysis of U$^3$CF and D-CGCT.

| Method | Mem Bank | Params (M) | Peak Mem (GB) | Train Time 500iters (s) | FLOPs (G) | Inference Latency (ms/image) | Inference Throughput (img/s) |
|---|---|---|---|---|---|---|---|
| D-CGCT (step1) | ✗ | 24.2 | 13.6 | 195 | 263.6 | 1.01 | 990.9 |
| D-CGCT (step2) | ✗ | 26.3 | 13.6 | 507 | 527.9 | 1.04 | 965.9 |
| **D-CGCT (step3)** | ✗ | 24.2 | 13.6 | 510 | 527.6 | 1.05 | 954.7 |
| U$^3$CF (step1-stage1) | ✓ | 29.1 | 13.1 | 130 | 261.8 | 0.89 | 1128.6 |
| U$^3$CF (step1-stage2) | ✓ | 29.1 | 13.1 | 275 | 524.0 | 0.93 | 1070.7 |
| **U$^3$CF (step2)** | ✓ | 29.1 | 13.1 | 293 | 528.1 | 0.98 | 1020.4 |

because the memory bank in U$^3$CF does not grow without bound. It only maintains a memory module with the same scale as the training samples, and for large-scale datasets such as DomainNet, we cap its size at 10,000. In terms of model size, U$^3$CF increases the peak parameter count by 2.8M.

② **Training-time cost.** U$^3$CF is consistently faster than D-CGCT during training. This shows that progressively expanding the memory bank and training dual classifiers do not introduce prohibitive training overhead.

③ **Computational complexity.** The FLOPs (floating-point operations) of U$^3$CF are lower than those of D-CGCT in the first two stages and only slightly higher in the third stage, indicating comparable overall computational complexity.

④ **Inference efficiency.** U$^3$CF remains more efficient at inference than D-CGCT, with consistently lower latency and higher throughput across all stages.

⑤ **Overhead of the causal strategy in Step 2.** To isolate the cost of the causal component, we further compare U$^3$CF Step 1-Stage 2 and Step 2. After introducing the causal strategy, the parameter count and peak memory remain unchanged, while the increases in training cost, computational complexity, and inference latency are all limited. This indicates that the causal strategy introduces only limited extra overhead.

Overall, although the memory module increases the peak parameter count by 2.8M over D-CGCT, it does not hurt efficiency. U$^3$CF remains faster in both training and inference, likely because it has a simpler network structure and does not adopt the CDAN architecture or the GRL strategy used in many MTDA methods. Moreover, the results show that the memory bank, dual classifiers, and causal strategy do not incur memory or efficiency overhead compared with standard MTDA baselines.

