# OpenReview forum: "U$^3$CF: Unbiased, Unconfounding, and Unified Causal Framework for Multi-Target Domain Adaptation"
_ICML.cc/2026/Conference — ICML 2026 regular_

### Official Review · Reviewer_7HsD · 2026-02-25

**Soundness:** 3
**Presentation:** 2
**Significance:** 3
**Originality:** 3
**Overall Recommendation:** 4
**Confidence:** 5

**Summary:**

This paper proposes U3CF, a causal framework for Multi-Target Domain Adaptation that addresses adaptation bias by explicitly modeling and removing domain-specific confounders. The method decomposes features into domain-invariant causal factors ($f_i$) and domain-specific confounding factors ($f_s$) via mutual information minimization, then applies conditional intervention (averaging predictions over domain prototypes $\{\hat{f}_{sd}\}$) to eliminate spurious correlations caused by $f_s$.

**Compliance With Llm Reviewing Policy:**

Affirmed.

**Final Justification:**

I hope the authors will carefully improve the writing and expand the discussion on the parameter $\lambda_{ali}$ in future versions. I maintain my original score.

**Key Questions For Authors:**

Please refer to Weaknesses.

**Strengths And Weaknesses:**

Strengths:

1. The way this work handles causal intervention is particularly novel. It adopts the principle of disentanglement and reconstruction, which differs from the causal intervention approaches I have encountered before. I learned a new methodology here, which leads me to consider the innovation level of this work to be quite significant.

Weaknesses:

1.  Poor readability: It took me nearly five times longer to understand this work compared to reviewing other papers. The paper proves several theoretical propositions that appear largely irrelevant to the core contribution, unnecessarily complicating the presentation. In my view, many of these proofs add little practical value.

2.  Notation clarity: The formulation of causal intervention in recent literature is typically expressed more cleanly as $ P(Y | do(X)) = \sum_{F_s} P(Y | F_i, F_s) P(F_s) $. This standard notation is significantly more intuitive and easier to follow than the authors' Equations (3) and (4).

3.  Conceptual ambiguity regarding confounders: On the right side of line 181, the authors claim that "$F_i$ is also a confounder." This seems counter-intuitive and potentially unreasonable. In most literature I have read, only the domain-specific factor $F_s$ is treated as the confounder, while $F_i$ represents the invariant causal mechanism.

4.  Inconsistent loss formulation: Equations (15)–(18) are essentially optimizations of the mutual information objective in Equation (12). However, it is unclear why the authors alternate between using Cross-Entropy loss and KL-divergence loss for different terms. A more unified explanation or derivation would improve clarity.

5.  Lack of ablation on classifier design: The final model relies solely on a Graph Convolutional Network (GCN) classifier. The paper lacks an ablation study comparing the GCN classifier with the MLP classifier (or an ensemble), making it difficult to assess the necessity and contribution of the graph-based component.

6.  Missing hyperparameter sensitivity analysis: The method involves several hyperparameters, yet the paper provides no discussion or analysis of the model's sensitivity to these choices.

7.  No computational efficiency discussion: Given the method's considerable complexity (multi-stage training, memory bank, causal intervention, dual classifiers, etc.), a discussion on training time, inference latency, and memory consumption is essential for practical assessment but is entirely absent.

8.  Outdated baselines: For a submission to ICML 2026, the experimental comparison is concerning. The authors only include baselines up to 2024, omitting more recent state-of-the-art methods in domain adaptation and causal representation learning published in 2025.

---

> ### Author Rebuttal · Authors · 2026-03-30
>
> **Dear Reviewer 7HsD,**
>
> Thank you for your constructive feedback. We have carefully considered all comments and sincerely appreciate the insightful questions. We hope our responses could resolve the concerns.
> >**Q1.** Poor readability & **Q2.** Notation clarity & **Q3.** Conceptual ambiguity regarding confounders
>
> **A1. & A2. & A3.** We thank the reviewer for these helpful comments. We agree that the current presentation can be improved in both readability and notation clarity. Regarding notation, we agree that Eqs. (3)–(4) can be presented more clearly.
>
> Our goal is to model both $F_i$ and $F_s$ as latent parent factors of $X$, where $F_i$ captures the stable semantic mechanism relevant to $Y$. The $F_i$ is better viewed as an invariant causal mechanism that should be preserved, whereas $F_s$ is the domain-specific confounder to be marginalized out by intervention. We will further refine these aspects in the revised version.
> >**Q4.** Inconsistent loss formulation
>
> **A4.** We thank the reviewer for the helpful comment. Eqs. (15)–(18) follow from Eq. (12), which includes both relevance and irrelevance mutual-information objectives. Proposition 4 provides different variational bounds for these two cases: Eq. (13) gives a lower bound for relevance, leading to CE loss for MI maximization, while Eq. (14) gives an upper bound for irrelevance, leading to KL loss for MI minimization. Thus, **CE and KL in Eqs. (15)–(18) are derived from Proposition 4, with the proof given in Appendix A.4.** We will further clarify this in the revised version.
> >**Q5.** Lack of ablation on classifier design
>
> **A5.** We thank the reviewer for the helpful comment. We separately ablate GCN and MLP in both the full U$^3$CF model and Step 2. The results show that removing GCN causes a larger drop than removing MLP, and the same trend is observed in Step 2. Overall, removing either GCN or MLP degrades performance, while GCN contributes more.
> Ablations on MTRS.|Avg.
> -|-
> U$^3$CF w/o GCN|77.2
> U$^3$CF w/o MLP|84.1
> U$^3$CF step 2 w/o GCN|86.7
> U$^3$CF step 2 w/o MLP|87.2
> >**Q6.** Missing hyperparameter sensitivity analysis
>
> **A6.** We thank the reviewer for the helpful comment. In Step 1-Stage 1, only $\lambda_{node}=0.3$ is introduced for datasets except DomainNet. In Step 1-Stage 2, only one additional parameter $\lambda_{ali}=0.1$ is introduced for all datasets. In Step 2, four parameters $\lambda_{dis}$, $\lambda_{rec}$, $\lambda_{ce}$, $\lambda_{kl}$ are introduced, and **their joint sensitivity is shown in Fig. 4 of the main paper**.
>
> Overall, most hyperparameters are the same across datasets, indicating good stability. The sensitivity analysis of $\theta$ is provided in Reviewer **F8Ys A1**. The sensitivity analysis of $\lambda_{ali}$ is shown below.
> $\lambda_{ali}$|0.1|0.2|0.3|0.4|0.5
> -|-|-|-|-|-
> MTRS|88.1|87.7|86.2|83.4|82.2
> >**Q7.** No computational efficiency discussion
>
> **A7.** We thank the reviewer for the helpful comment. We provide a detailed overhead analysis of U$^3$CF. Although the memory module increases the peak parameter count by 2.8M over D-CGCT, it does not reduce efficiency. Since all stages include the memory bank and dual classifiers, and Step 2 further introduces the causal strategy, we analyze the overhead of each stage separately. More details are provided in Reviewer **2ha6 A2**.
> Method|Mem Bank|Params (M)|Peak Mem (GB)|Train Time 500iters (s)|FLOPs (G)|Inference Latency (ms/image)|Inference Throughput (img/s)
> -|-|-|-|-|-|-|-
> D-CGCT (step1)|✗|24.2|13.6|195|263.6|1.01|990.9
> D-CGCT (step2)|✗|26.3|13.6|507|527.9|1.04|965.9
> **D-CGCT (step3)**|✗|24.2|13.6|510|527.6|1.05|954.7
> U$^3$CF (step1-stage1)|✓|29.1|13.1|130|261.8|0.89|1128.6
> U$^3$CF (step1-stage2)|✓|29.1|13.1|275|524.0|0.93|1070.7
> **U$^3$CF (step2)**|✓|29.1|13.1|293|528.1|0.98|1020.4
> >**Q8.** Outdated baselines
>
> **A8.** We further compare with 2025 baselines. Since MTDA remains relatively underexplored, recent work mainly considers related settings such as MSMTDA (Multi-Source Multi-Target).
>
> We compare with the following methods.
>
> MSMTDA:
>
> MAN (TMM2025): Multiple adaptation network for multi-source and multi-target domain adaptation
>
> IPCA (AAAI2025): Invertible projection and conditional alignment for multi-source blended-target domain adaptation
>
> Vit-based MTDA:
>
> CACM (TCSVT2025)
>
> On OfficeHome under MSMTDA, **U$^3$CF clearly surpasses 2025 baselines MAN and IPCA by 4.9% and 4.4%, respectively**.
>
> On DomainNet, U$^3$CF is less competitive with DeiT-B but shows clear gains with CLIP-ViT-B and DINOv3-ViT-B at the same parameter scale (86M).
> Source→Target (OfficeHome)|R+P→A+C|C+R→A+P|P+C→A+R|R+A→C+P|A+P→C+R|C+A→P+R|Avg
> -|-|-|-|-|-|-|-
> MAN|68.4|77.3|72.1|72.9|71.5|77.1|73.2
> IPCA|65.6|79.9|75.2|70.1|71.8|79.3|73.7
> U$^3$CF|69.8|82.8|79.9|75.4|75.1|85.5|78.1
>
> Task (DomainNet)|C|I|P|Q|R|S|Avg
> -|-|-|-|-|-|-|-
> CACM (DeiT-B)|48.1|48.0|45.4|29.6|47.8|50.5|44.9
> U$^3$CF (CLIP-ViT-B)|54.8|55.1|50.9|21.2|57.7|56.6|49.4
> U$^3$CF (DINOv3-ViT-B)|56.1|55.5|52.7|35.9|59.1|57.9|52.9

---

> > ### Author Rebuttal · Reviewer_7HsD · 2026-04-03
> >
> > Dear Authors,
> >
> > Thank you for the thorough, detailed, and constructive rebuttal. I sincerely appreciate the extensive effort you put into addressing my concerns, particularly the inclusion of the recent baselines (Q8), the comprehensive computational efficiency analysis (Q7) and ablation on classifier design (Q5).
> >
> > However, after carefully reviewing your responses, I still have two lingering reservations:
> >
> > 1. Presentation and Readability (Q1-Q3)
> >
> > The current writing, notation, and conceptual explanations (e.g., regarding confounders) remain overly convoluted, which severely hampers the paper's readability. While I appreciate your commitment to revising the notations and streamlining the presentation in the final version, I can only evaluate the manuscript currently in front of me.
> >
> > 2. Hyperparameter Sensitivity (Q6)
> >
> > Regarding the sensitivity analysis for $\lambda_{ali}$, you mentioned that the method exhibits "good stability." However, the provided results show a performance drop from 88.1 (at $\lambda_{ali} = 0.1$) to 82.2 (at $\lambda_{ali} = 0.5$). A degradation of nearly 6% within this range suggests that the model is, in fact, quite sensitive to this specific hyperparameter, which raises practical concerns about tuning and robustness.
> >
> > Overall, I still hold a positive view of this work. The causal intervention approach is novel, and your rebuttal successfully resolved my concerns regarding the baselines and efficiency.
> > Nevertheless, considering the significant presentation issues and the model's sensitivity to hyperparameter selection, I believe my current score accurately reflects the paper's standing. Therefore, I will maintain my original score of 4 (Weak Accept).

---

> > > ### Author Response · Authors · 2026-04-03
> > >
> > > **Dear Reviewer 7HsD,**
> > >
> > > Thank you for your feedback. We hope that the responses below help clarify our points and address your concerns.
> > >
> > > >**Q1. The current writing, notation, and conceptual explanations (e.g., regarding confounders) remain overly convoluted.**
> > >
> > > **A1.** Thank you for this question.
> > >
> > > **① Conceptual explanation of the confounder**
> > >
> > > In our SCM (Fig. 3), $F_i$ influences both $X$ and $Y$, and therefore can be viewed as a confounder. Under Pearl’s causal-graph framework, $F_i$ can be interpreted as a confounder for the effect of $X$ on $Y$, because it points to both $X$ and $Y$ and therefore induces the back-door path $X \leftarrow F_i \rightarrow Y$ [1].
> > >
> > > Based on the above analysis, our previous characterization of $F_i$ as a confounder is theoretically justified.
> > >
> > > As a domain-invariant feature, $F_i$ can also be viewed as an invariant causal mechanism, as adopted in prior work. The key point is whether it affects both $X$ and $Y$. For clarity, we will further refer to $F_i$ as an invariant causal mechanism.
> > >
> > > **References:**
> > >
> > > [1] Pearl, J. (2009). *Causality: Models, Reasoning, and Inference* (2nd ed.). Cambridge University Press.
> > >
> > > **② Overly convoluted**
> > >
> > > **Our original intention was to help readers better understand the model details. The derivations of Eqs. (3) and (4) are correct, and we presented them in full form to make clear how each term is defined and specified**.
> > >
> > > We thank the reviewer for the helpful suggestion. In the revised version, we will simplify the presentation throughout the main paper and provide the complete formulation in the appendix, so that readers can gain both a concise and a comprehensive understanding of the method.
> > >
> > > >**Q2. Hyperparameter Sensitivity (Q6).**
> > >
> > > **A2.** Thank you for this question.
> > >
> > > ① Hyperparameter sensitivity is an important issue for machine learning methods. **We would like to emphasize that sensitivity to a specific hyperparameter does not necessarily mean that the overall method lacks robustness. In our experiments, we fixed $\lambda_{ali}=0.1$ across all datasets and still obtained the best average accuracy across all evaluated datasets**. This suggests that the parameter admits a practical and effective choice.
> > >
> > > ② Furthermore, **except for DomainNet, we used the same hyperparameter configuration for all other datasets, while still achieving leading performance across all evaluated datasets. This shows that our method does not depend on extensive dataset-specific tuning, while still maintaining stable performance**. From this perspective, we believe the method maintains good practical robustness overall.
> > >
> > > *In the revised version, we will revise the manuscript accordingly and incorporate the corresponding clarifications based on your suggestions.*
> > >
> > > *We sincerely appreciate your comments and would be grateful for your reconsideration of our responses. Thank you again for your time and effort.*
> > >
> > > Best regards,
> > >
> > > Authors of #10630

---

### Official Review · Reviewer_fMWh · 2026-03-05

**Soundness:** 3
**Presentation:** 3
**Significance:** 2
**Originality:** 2
**Overall Recommendation:** 4
**Confidence:** 4

**Summary:**

The paper proposes a structure for unbiased prediction in MTDA which uses prototype-drvien alignment strategy and formulates a structural causal model for revealing domain-invariant factors.
For unbiased prediction, it disentangles representations into invariant causal components and domain-specific confounders and blocking them using intervention.

**Compliance With Llm Reviewing Policy:**

Affirmed.

**Final Justification:**

Most of my concerns have been addressed. I will maintain my score.

**Key Questions For Authors:**

1. What assumption makes the uniform domain averaging to unbiased estimator? Please explain both estimand and estimator.
2.  Additional empirical evidences that can show the independence assumptions on domain-specific features and domain-invariant features that can support your theoretical analysis.
3.  More evidences for choosing uniform prior and prototype averaging.
4.  Are there any clues that confounders can be represented by the prototypes? What happens when $K_D$ becomes larger or domain becomes more unbalnced?

**Limitations:**

More experimental evidences are required to support their ideas which can directly explain their hypothesis and designs.

**Strengths And Weaknesses:**

Strengths
1. The paper provides abundant experimental evidences using four benchmarks.
2. The paper is motivated well that domain-specific factor causes spurious correlation in MTDA and it is quite well-structured that address this issue with SCM and confounders.
---
Weaknesses
1. It seems that the claims on 'unbiased' and 'causal' are quite weak. The paper really proposes SCM and do-intervention theoretically but it is unclear under what explicit assumptions the proposed computational procedure can be interpreted as an identified estimator for domain-invariant prediction. Especially, $F_s$ seems that it is just mean of the possible confounders so calling it as 'unbiased' at this stage represents a logical leap in the argument.
2. More explanations are required for $P(F_s|F_i)=P(F_s)$ which you used to model them as independent parent factors. In real scenarios, this assumption is brittle since it is hard to disentangle domain-specific features and domain-invariant features. It needs more empirical evidences that can show your design works.
3. The methods relies on the $K_D$. It needs more explanations for prototypes, uniform prior and domain-averaged intervention. And correspondence to reality in both theoretical and empirical methods.

---

> ### Author Rebuttal · Authors · 2026-03-31
>
> **Dear Reviewer fMWh,**
>
> Thank you for your constructive feedback. We have carefully considered all comments and sincerely appreciate the insightful questions. We hope our responses could resolve the concerns.
> >**Q1.** Estimand and Estimator
>
> **A1.** We thank the reviewer for the helpful comment. The interventional **estimand** is Eq. (3), i.e., the conditional intervention that preserves $F_i$ while marginalizing the domain-specific confounder $F_s$.
>
> Eq. (4) is a **sample approximation estimator** of this estimand, obtained by approximating $F_s$ with $K_D$ domain prototypes and assigning uniform weights to these confounders.
>
> Under this interpretation, the current estimator is “unbiased” in the sense of **(i) Treat each domain equally** and **(ii) Ignore domain identity**. However, our claim is not that the estimator is guaranteed to be statistically unbiased for any real-world distribution.
> - **(i) Treat each domain equally**: each domain contributes equally to the prediction, regardless of its empirical frequency.
> - **(ii) Ignore domain identity**: $g({f}_i)$ in Proposition 2 is implicitly obtained from $G_r$. In Eq. (21), each sample is reconstructed with all $K_D$ domain prototypes, regardless of its domain label, thereby reducing the effect of domain identity.
>
> The assumptions are:
> - (1) the domain labels are known, so the domain-specific confounder can be estimated.
> - (2) the domain-specific confounder can be approximated by domain prototypes.
>
> >**Q2.** Independence assumptions
>
> **A2.** We thank the reviewer for the helpful comment.
>
> **(1) Theoretical level:** The independence assumption is a standard factorized variational approximation widely used in VAEs and disentanglement-style modeling.
>
> A similar theory is used in the reference **"Variational Interaction Information Maximization for Cross-Domain Disentanglement"** cited in the main paper.
>
> **(2) Experimental level:** We further validate the independence assumption experimentally. Specifically, we quantitatively evaluate the disentanglement between $f_i$ and $f_s$, and the classifiers in the FDM (Fig. 2) provide such evidence. On Office-31 (W), $f_i$ achieves high accuracy on the category classifier $C_c$ but is not discriminative for the domain classifier $C_d$, whereas $f_s$ shows the opposite behavior. The cosine similarity between $f_i$ and $f_s$ is only 0.09, indicating good disentanglement.
> Office-31 (W)|Class Acc|Domain Acc
> -|-|-
> $f_i$|85.6|37.2
> $f_s$|5.2|97.9
> $y_{causal}$|88.6|\
> >**Q3.** Uniform prior and prototype averaging & **Q4.** Confounder and imbalanced domain
>
> **A3. & A4.** We thank the reviewer for these helpful comments.
> ### **(1) Uniform Prior**
> We aim to treat each domain equally and therefore adopt a uniform prior. Proposition 1 and Appendix A.1 provide the theoretical proof, further supported by the experiments below.
>
> **① Imbalanced-Domain Experiments**
>
> We compare uniform prior and empirical prior (i.e., dynamic weighting), and find that their performance is related to the evaluation metric. Our metric (Eq. (64)) is Macro Avg Accuracy, rather than Micro Avg Accuracy. We compare **Uniform prior** and **Empirical prior** on MTRS and an imbalanced MTRS* (P,R,U,A,N=9600:4800:2400:1200:300=32:16:8:4:1).
>
> We report **Macro Avg Accuracy** (equal-weight across domains) and **Micro Avg Accuracy** (sample-weighted across domains). On MTRS, Uniform prior improves Macro Avg by 0.6\%, while Empirical prior improves Micro Avg by 0.3\%. On MTRS*, Uniform prior improves Macro Avg by 5.9\%, while Empirical prior improves Micro Avg by 0.7\%. This means that Uniform prior works better for Macro Avg, while Empirical prior works better for Micro Avg.
> Data|Macro Avg. (Uniform)|Macro Avg. (Empirical)|Micro Avg. (Uniform)|Micro Avg. (Empirical)
> -|-|-|-|-
> MTRS|88.1|87.5|85.7|86.0
> MTRS*|74.6|68.7|69.4|70.1
> ### **(2) Prototype and Confounder**
> Domain prototypes are not assumed to be exact confounders, but serve as a sample approximation of the domain-specific confounder in Eq. (4). We further validate this with experiments on different confounder strategies.
>
> **② Confounder Strategy Experiments**
>
> We compare a mean-based domain prototype strategy with a confidence-based top-1 confounder strategy. As shown below, the domain prototype strategy adopted in our method performs better.
> MTRS|Avg.
> -|-
> confidence-based confounder|86.7
> prototype-based confounder (ours)|88.1
>
> **③ Noise Robustness Experiments**
>
> We add random noise to the Office-31 (W). The accuracy of $f_i$ on $C_c$ drops by only 6.4\%, while the accuracy of $f_s$ on $C_d$ drops by 20.7\%. **This suggests that U$^3$CF captures domain-specific confounding factors in $f_s$, so noise affects $f_s$ more while preserving the class-discriminative information in $f_i$, validating the effectiveness of the confounder design.**
> Office-31 (W)|Class Acc|Domain Acc
> -|-|-
> $f_i$|85.6|37.2
> $f_s$|5.2|**97.9**
> $y_{causal}$|88.6|\
> $f_i(noise)$|79.2|34.5
> $f_s(noise)$|4.1|**77.2**
> $y_{causal}(noise)$|82.7|\

---

> > ### Author Rebuttal · Reviewer_fMWh · 2026-04-02
> >
> > This rebuttal shows that this method is the empirically competent MTDA method that borrows causal language for equal-domain macro prediction.
> > Several questions are remained:
> > 1. After your clarification, what formal guarantee does Eq. (4) have for approximating Eq. (3), beyond equal-domain averaging?
> > 2. How do you justify SCM-level independence between domain-specific features and domain-invariant features, rather than only encoder factorization and low cosine similarity?
> > 3. Should the paper explicitly state that the uniform prior is a modeling choice aligned with the macro-average metric, rather than a consequence of the SCM?
> > 4. What evidence shows that domain prototypes approximate confounders rather than merely serving as useful heuristics, especially under more domains or stronger imbalance?

---

> > > ### Author Response · Authors · 2026-04-03
> > >
> > > **Dear Reviewer fMWh,**
> > >
> > > We appreciate your follow-up questions and hope our response addresses your concerns.
> > >
> > > >Q1
> > >
> > > **A1.** Thank you for this question.
> > >
> > > By specifying $F_s$ and $P(F_s)$, Eq. (4) is obtained as a concrete instantiation of Eq. (3).
> > >
> > > Eq. (3) is the general interventional expression derived by applying a conditional intervention to our SCM for MTDA (Fig. 3). This intervention is theoretically grounded in intervention theory [1].
> > >
> > > In our previous rebuttal, we referred to Eq. (4) as an approximation for two reasons. First, the confounder is difficult to compute directly, so we use domain prototypes to approximate $F_s$. Second, we model $P(F_s)$ using a uniform prior.
> > >
> > > [1] A Calculus for Stochastic Interventions: Causal Effect Identification and Surrogate Experiments (AAAI2020)
> > > >Q2
> > >
> > > **A2.** Thank you for this question.
> > >
> > > In our method, the independence between domain-specific features and domain-invariant features at the SCM level is introduced as a structural modeling assumption, motivated by the MTDA setting.
> > >
> > > This is common in SCM-based causal learning methods, where the causal structure is specified as a modeling assumption or structural prior, rather than used to establish SCM-level independence [2, 3].
> > >
> > > By contrast, encoder factorization and low cosine similarity provide only **representation-level evidence** consistent with this assumption. This assumption guides the SCM construction, while the training objectives encourage the disentanglement and the experiments provide supporting evidence.
> > >
> > > [2] Weakly supervised causal representation learning (2022NeurIPS)
> > >
> > > [3] Variational Interaction Information Maximization for Cross-domain Disentanglement (2020NeurIPS)
> > >
> > > We will clarify this assumption in the revised version.
> > > >Q3
> > >
> > > **A3.** Thank you for this question.
> > >
> > > We do not claim the uniform prior as a consequence of the SCM, and we apologize for any confusion caused by our explanation.
> > >
> > > What the SCM provides is Eq. (3), which serves as the guiding formulation for the subsequent learning procedure. Eq. (3) requires a prior over $P(F_s)$, but the choice of $P(F_s)$ is not determined by the SCM and can be specified based on the practical setting.
> > >
> > > Through the rebuttal experiments, we find that prior selection may be related to the evaluation metric. In the general MTDA setting, the uniform prior is more suitable to achieve better overall performance.
> > >
> > > **Therefore, the role of our SCM and conditional intervention is to provide a principled learning framework, which $P(F_s)$ can be flexibly specified according to the practical scenario. This flexibility makes the framework more extensible and better suited to real-world dynamic environments**.
> > >
> > > We will clarify in the revised version that the uniform prior is more appropriate for the Micro Avg.
> > > >Q4
> > >
> > > **A4.** Thank you for this question.
> > >
> > > ### **(1) domain prototypes approximate confounders**
> > > ① In the SCM, the domain-specific feature $f_s$ acts as the confounder. Each domain prototype is constructed from the $f_s$ of samples in the same domain.
> > >
> > > ② **Our method aims to learn the $f_s$ for each sample as accurately as possible and uses their mean as the confounder**. Since confounder construction is not unique, we compared our strategy (88.1%) with an alternative that uses the $f_s$ of the top-1 confidence sample as the confounder (86.7%). The results support our strategy.
> > >
> > > ③ Classifier results and cosine similarity suggest relative independence between $f_s$ and $f_i$. Moreover, $f_s$ is domain-discriminative but not class-discriminative, indicating that it captures class-irrelevant factors, such as domain-specific environmental characteristics. Eqs. (15)–(18) in the paper and the last table in the rebuttal also support this point.
> > >
> > > ④ We also want to highlight the Noise Robustness Experiments in the rebuttal. Under the same random noise across domains, the domain-invariant features remain largely stable, while the domain-specific features are affected much more. This suggests that U$^3$CF mainly captures confounding factors in $f_s$ to construct the confounder, further supporting the effectiveness of our domain prototype strategy.
> > > ### **(2) more domains or stronger imbalance**
> > > Whether domain prototypes can approximate confounders is not determined by the number of domains. As explained above, each domain prototype is constructed only from the $f_s$. Therefore, increasing the number of domains does not affect the prototype of any individual domain.
> > >
> > > For imbalanced domains, too few samples may result in insufficient prototype learning. Meanwhile, severe data imbalance commonly has a negative impact on model performance. U$^3$CF is designed for general MTDA rather than extreme imbalance.
> > >
> > > We constructed the "more domain, stronger imbalanced" MTRS* dataset, which has 5 domains and an imbalance ratio of 32:16:8:4:1. Compared with D-CGCT, U$^3$CF achieves a 5.1% gain, validating its effectiveness on this type of data.
> > > MTRS*|Avg.
> > > -|-
> > > U$^3$CF|74.6
> > > D-CGCT|69.5

---

### Official Review · Reviewer_F8Ys · 2026-03-11

**Soundness:** 3
**Presentation:** 3
**Significance:** 3
**Originality:** 3
**Overall Recommendation:** 4
**Confidence:** 4

**Summary:**

In the paper, the authors propose the Unbiased, Unconfounding, and Unified Causal Framework (U 3CF) for multi-target domain adaptation (MTDA). A prototype-driven alignment strategy is designed to progressively update prototypes by high-confidence target predictions. A structural causal model is formulated to reveal that domain-invariant causal factors.

**Compliance With Llm Reviewing Policy:**

Affirmed.

**Key Questions For Authors:**

1.	How sensitive is the framework to the choice of the pseudo-label confidence threshold $\theta$? Is there a risk of confirmation bias accumulating in the memory bank $\mathcal{M}$ during the early stages of training if the initial domain shift is severe, and how does the model mitigate this?
2.	In Proposition 2 and Eq. (4), the conditional intervention uses a uniform prior for domain prototypes ($1/K_D$). How does the framework perform when target domains are highly imbalanced in terms of sample size or label distribution? Would a dynamically weighted intervention be more robust?
3.	The method relies heavily on the GCN classifier architecture from the D-CGCT baseline. Could you clarify how much of the performance gain is strictly attributable to the causal disentanglement intervention versus the graph-based message passing? Have you tested the causal classification branch solely with the MLP classifier?
4.	What is the additional computational and memory overhead introduced by the progressive memory bank expansion and the mutual information-based disentanglement module compared to standard MTDA baselines?
5.	While the t-SNE visualizations are helpful, are there quantitative metrics you can provide to explicitly measure the degree of disentanglement achieved between the invariant features $f_i$ and domain-specific features $f_s$?

**Limitations:**

The authors have not adequately discussed the limitations and potential negative societal impacts of their work. The paper includes a brief “Impact Statement” (Lines 440-445), but it dismisses the need to discuss societal consequences. Furthermore, there is no dedicated discussion regarding the technical limitations of the proposed method.

**Strengths And Weaknesses:**

Strengths
1.	The paper is not merely a simple empirical improvement, but constructs a Structural Causal Model (SCM) to identify domain-specific factors in MTDA as “confounding factors” and provides four rigorous proofs of propositions. For example, Proposition 1 proves that alignment loss is a lower bound of mutual information, and Proposition 2 proves the unbiasedness of conditional intervention, significantly enhancing the academic depth of the work.
2.	The framework successfully decomposes the representation into domain-invariant features ($F_i$) and domain-specific features ($F_s$) and utilizes mutual information minimization to ensure the purity of decoupling. This method, which blocks the confounding path through “soft intervention” (i.e., averaging on domain prototypes), is highly innovative in multi-target domain adaptation scenarios.
3.	The experiment encompassed four mainstream benchmarks, including the highly challenging DomainNet. The results indicated that U^3CF outperformed existing state-of-the-art (SOTA) methods, such as MCDA and D-CGCT, in terms of average accuracy across all datasets. Specifically, on the MTRS remote sensing dataset, it achieved a 1.9% improvement over the second-best method.
Weaknesses
1.	The framework comprises two primary training steps and multiple intricate loss functions, encompassing approximately 10 distinct weight parameters. Despite its exceptional performance, the two-stage training process and feature decoupling module notably augment training duration and memory usage. The paper lacks a comparative analysis regarding inference speed or computational efficiency.
2.	Sensitivity analysis reveals that performance is significantly influenced by multiple coefficients such as $\lambda_{dis}, \lambda_{rec}, \lambda_{ce}, \lambda_{kl}$. For instance, $\lambda_{dis}$ needs to be maintained within a narrow range of [0.2, 0.4] to achieve optimal results. When deploying to new tasks, this complex multi-parameter tuning may incur significant search costs.
3.	The causal model assumes that $F_s$ and $F_i$ are independent parent factors. However, in actual complex visual scenes, domain-specific confounding factors (such as lighting or background) often exhibit nonlinear coupling with the causal features of the object itself (such as shape). Simple decoupling may not fully disentangle these mutual influences, and the discussion of such extreme failure cases in the paper is somewhat insufficient.

---

> ### Author Rebuttal · Authors · 2026-03-30
>
> **Dear Reviewer F8Ys,**
>
> Thank you for your constructive feedback. We have carefully considered all comments and sincerely appreciate the insightful questions. We hope our responses could resolve the concerns.
> >**Q1.** Confirmation bias.
>
> **A1.** We thank the reviewer for the helpful comment. The parameter $\theta$ is used to select high-confidence target samples, reducing error accumulation in the memory bank. As described in Appendix D.2, we set $\theta=0.85$ for DomainNet and $\theta=0.8$ for all other datasets. $\theta$ is chosen to keep the accuracy of selected samples above 80\%. However, a larger $\theta$ yields fewer selected target samples and may hurt cross-domain generalization.
>
> **Pseudo-label noise is inevitable in domain adaptation, but its accumulation is controlled in our framework**. In Step 1-Stage 1, $\mathcal{M}$ is initialized only with labeled source samples, introducing no pseudo-label error. In Step 1-Stage 2, only high-confidence target samples are added to memory bank, limiting the accumulated error. In Step 2, $\mathcal{M}$ is fixed, so no further error is introduced.
> $\theta$|0.6|0.7|0.75|0.8|0.85|0.9
> -|-|-|-|-|-|-
> Sensitivity analysis on MTRS|81.2|84.8|86.9|88.1|88.2|87.5
> >**Q2.** Dynamically intervention.
>
> **A2.** We thank the reviewer for the helpful comment. Dynamically weighted intervention only changes $P(F_s)$ in Eq. (3). Its benefit depends on the evaluation metric. Since our main metric (Eq. (64)) is Macro Avg Accuracy, rather than Micro Avg Accuracy. We compare **Uniform prior** and **Empirical prior** on MTRS and a **highly imbalanced MTRS***(P,R,U,A,N=9600:4800:2400:1200:300=32:16:8:4:1).
>
> We report **Macro Avg Accuracy** (equal-weight across domains) and **Micro Avg Accuracy** (sample-weighted across domains). On MTRS, Uniform prior improves Macro Avg by 0.6\%, while Empirical prior improves Micro Avg by 0.3\%. On MTRS*, Uniform prior improves Macro Avg by 5.9\%, while Empirical prior improves Micro Avg by 0.7\%.
>
> **This means that Uniform prior works better for Macro Avg, while Empirical prior works better for Micro Avg**.
> Data|Macro Avg. (Uniform)|Macro Avg. (Empirical)|Micro Avg. (Uniform)|Micro Avg. (Empirical)
> -|-|-|-|-
> MTRS|88.1|87.5|85.7|86.0
> MTRS*|74.6|68.7|69.4|70.1
> >**Q3.** Ablations.
>
> **A3.** We thank the reviewer for the helpful comment. We ablate GCN and MLP in both the full U$^3$CF model and Step 2. The results show that removing GCN causes a larger drop than removing MLP, and the same trend is observed in Step 2. Overall, removing either GCN or MLP degrades performance, while GCN contributes more.
>
> Together with **the table below and Table 5 in the main paper**, these results show that the causal strategy in Step 2 improves Avg. by 1.7\%. Without the causal strategy, even joint MLP+GCN training still yields a 1.4% drop, **confirming that the causal design is the most important component in Step 2**.
> Ablations on MTRS|Avg.
> -|-
> U$^3$CF w/o GCN|77.2
> U$^3$CF w/o MLP|84.1
> U$^3$CF step 2 w/o GCN|86.7
> U$^3$CF step 2 w/o MLP|87.2
> U$^3$CF step 2 w/o Causal|86.7
> U$^3$CF-3|86.4
> U$^3$CF|**88.1**
> >**Q4.** Overhead.
>
> **A4.** We thank the reviewer for the helpful comment. We provide a detailed overhead analysis of U$^3$CF. Although the memory module increases the peak parameter count by 2.8M over D-CGCT, it does not reduce efficiency. U$^3$CF remains faster in both training and inference, likely because it does not use the CDAN architecture or a GRL layer. More details are provided in Reviewer **2ha6 A2**.
> |Method|Mem Bank|Params (M)|Peak Mem (GB)|Train Time 500iters (s)|FLOPs (G)|Inference Latency (ms/image)|Inference Throughput (img/s)|
> |-|-|-|-|-|-|-|-|
> |D-CGCT (step1)|✗|24.2|13.6|195|263.6|1.01|990.9
> |D-CGCT (step2)|✗|26.3|13.6|507|527.9|1.04|965.9
> |**D-CGCT (step3)**|✗|24.2|13.6|510|527.6|1.05|954.7
> |U$^3$CF (step1-stage1)|✓|29.1|13.1|130|261.8|0.89|1128.6
> |U$^3$CF (step1-stage2)|✓|29.1|13.1|275|524.0|0.93|1070.7
> |**U$^3$CF (step2)**|✓|29.1|13.1|293|528.1|0.98|1020.4
> >**Q5.** Quantitative metrics.
>
> **A5.** We thank the reviewer for the helpful comment. We quantify disentanglement using **Class Acc, Domain Acc, and cosine similarity**, as provided by the classifiers in the FDM (Fig. 2). On Office-31 (Webcam), $f_i$ achieves high accuracy on the category classifier $C_c$ but is not discriminative for the domain classifier $C_d$, whereas $f_s$ shows the opposite behavior. The cosine similarity between $f_i$ and $f_s$ is only 0.09, **indicating good disentanglement**.
>
> We further add **random noise** to the data. The accuracy of $f_i$ on $C_c$ drops by only 6.4\%, while the accuracy of $f_s$ on $C_d$ drops by 20.7\%. **This suggests that U$^3$CF primarily captures confounding factors in $f_s$, so noise perturbs $f_s$ more while preserving the class-discriminative information in $f_i$**.
> Office-31 (Webcam)|Class Acc|Domain Acc
> -|-|-
> $f_i$|85.6|37.2
> $f_s$|5.2|**97.9**
> $y_{causal}$|88.6|\
> $f_i (noise)$|79.2|34.5
> $f_s (noise)$|4.1|**77.2**
> $y_{causal} (noise)$|82.7|\

---

> > ### Author Rebuttal · Reviewer_F8Ys · 2026-04-05
> >
> > Thank you for the rebuttal. All the questions have been addressed, thus, I keep my positive score.

---

> > > ### Author Response · Authors · 2026-04-05
> > >
> > > **Dear Reviewer F8Ys,**
> > >
> > > Thank you very much for your time and thoughtful consideration of our rebuttal.
> > >
> > > We sincerely appreciate your acknowledgment that our responses have addressed your concerns.
> > >
> > > In the revised version, we will revise the manuscript accordingly and incorporate the corresponding clarifications based on your suggestions.
> > >
> > > If you find the clarifications and additional evidence helpful, we would be grateful if they could be considered in your final evaluation.
> > >
> > > Best regards,
> > >
> > > Authors of #10630

---

### Official Review · Reviewer_2ha6 · 2026-03-12

**Soundness:** 3
**Presentation:** 3
**Significance:** 3
**Originality:** 3
**Overall Recommendation:** 4
**Confidence:** 2

**Summary:**

This paper introduces $U^3CF$, an Unbiased, Unconfounding, and Unified Causal Framework for Multi-Target Domain Adaptation (MTDA). The authors propose a two-stage approach: a progressive prototype-guided cross-domain alignment to handle heterogeneous shifts , followed by causal adaptation formulated via a Structural Causal Model (SCM). By utilizing mutual information theory to disentangle domain-invariant causal features from domain-specific confounders , the framework applies conditional intervention to block confounding effects, aiming for unbiased predictions across multiple target domains.

**Compliance With Llm Reviewing Policy:**

Affirmed.

**Final Justification:**

I have read the authors' responses to us and the other reviewers, and most of my questions have been resolved. Good Luck!

**Key Questions For Authors:**

1. The proposed two-stage training pipeline is highly complex and introduces a substantial number of hyperparameters (e.g., $\lambda_{node}, \lambda_{ali}, \lambda_{dis}, \lambda_{rec}, \lambda_{ce}, \lambda_{kl}$). This reliance on hyperparameter tuning diminishes the framework's practical usability and robustness.
2. Progressively expanding and updating the memory bank M with high-confidence target samples , alongside training dual classifiers (an MLP classifier and a GCN classifier), likely incurs significant memory footprint and training-time costs compared to standard MTDA baselines. The paper currently lacks a detailed analysis of computational complexity, training latency, and memory overhead.
3.  There are minor formatting and typographical inconsistencies throughout the mathematical formulations. For instance, some equations lack proper trailing punctuation.
4. There is a discrepancy between the architectural diagrams and the main text. Specifically, Figure 2 labels the module as a "GNN Classifier" , whereas the main text consistently refers to it as a "GCN classifier". This nomenclature should be standardized for clarity.

**Limitations:**

yes

**Strengths And Weaknesses:**

Strengths:

1. The paper provides a novel causal perspective for the MTDA task by explicitly identifying and modeling domain-specific factors as confounders within an SCM.
2. The theoretical grounding of the paper is robust.

Weaknesses:

1. The proposed two-stage training pipeline is highly complex and introduces a substantial number of hyperparameters (e.g., $\lambda_{node}, \lambda_{ali}, \lambda_{dis}, \lambda_{rec}, \lambda_{ce}, \lambda_{kl}$). This reliance on hyperparameter tuning diminishes the framework's practical usability and robustness.
2. Progressively expanding and updating the memory bank M with high-confidence target samples , alongside training dual classifiers (an MLP classifier and a GCN classifier), likely incurs significant memory footprint and training-time costs compared to standard MTDA baselines. The paper currently lacks a detailed analysis of computational complexity, training latency, and memory overhead.
3.  There are minor formatting and typographical inconsistencies throughout the mathematical formulations. For instance, some equations lack proper trailing punctuation.
4. There is a discrepancy between the architectural diagrams and the main text. Specifically, Figure 2 labels the module as a "GNN Classifier" , whereas the main text consistently refers to it as a "GCN classifier". This nomenclature should be standardized for clarity.

---

> ### Author Rebuttal · Authors · 2026-03-31
>
> **Dear Reviewer 2ha6,**
>
> Thank you for your constructive feedback. We have carefully considered all comments and sincerely appreciate the insightful questions. We hope our responses could resolve the concerns.
> >**Q1.** Hyperparameter Sensitivity and Robustness.
>
> **A1.** We appreciate the reviewer for the helpful comment. **Although our method includes several hyperparameters, they are introduced progressively across stages rather than all at once. Moreover, most of them share the same settings across datasets, which supports the stability of our method**. Specifically:
>
> In **Step 1-Stage 1**, only $\lambda_{node}$ is introduced, set to 0.5 for DomainNet and 0.3 for all other datasets.
>
> In **Step 1-Stage 2**, only one additional parameter, $\lambda_{ali}=0.1$, is introduced for all datasets.
>
> In **Step 2**, four parameters, $\lambda_{dis}$, $\lambda_{rec}$, $\lambda_{ce}$, and $\lambda_{kl}$, are introduced, and their joint sensitivity is shown in Fig. 4 of the main paper, where good performance is achieved within a reasonable range.
>
> We further provide sensitivity analyses of $\lambda_{ali}$ and $\theta$ below. For $\theta$, we set $\theta=0.85$ for DomainNet and $\theta=0.8$ for all other datasets.
>
> Overall, the hyperparameters are introduced progressively, and most of them are shared across datasets, indicating good stability.
> $\lambda_{ali}$|0.1|0.2|0.3|0.4|0.5
> -|-|-|-|-|-
> Sensitivity on MTRS|88.1|87.7|86.2|83.4|82.2
>
> $\theta$|0.6|0.7|0.75|0.8|0.85|0.9
> -|-|-|-|-|-|-
> Sensitivity on MTRS|81.2|84.8|86.9|88.1|88.2|87.5
> >**Q2.** Overhead of U$^3$CF.
>
> **A2.** We appreciate the reviewer for the helpful comment.
>
> We provide a detailed stage-wise overhead analysis of U$^3$CF. Specifically:
>
> **① Memory footprint and parameter count.** Despite introducing the memory bank and dual classifiers, U$^3$CF does not increase peak memory usage. Its peak GPU memory is consistently lower than that of D-CGCT across all stages. **This is because the memory bank in U$^3$CF does not grow without bound. It only maintains a memory module with the same scale as the training samples, and for large-scale datasets such as DomainNet, we cap its size at 10,000**. In terms of model size, U$^3$CF increases the peak parameter count by 2.8M.
>
> **② Training-time cost.** U$^3$CF is consistently faster than D-CGCT during training. This shows that progressively expanding the memory bank and training dual classifiers do not introduce prohibitive training overhead.
>
> **③ Computational complexity.** The FLOPs (floating-point operations) of U$^3$CF are lower than those of D-CGCT in the first two stages and only slightly higher in the third stage, indicating comparable overall computational complexity.
>
> **④ Inference efficiency.** U$^3$CF remains more efficient at inference than D-CGCT, with consistently lower latency and higher throughput across all stages.
>
> **⑤ Overhead of the causal strategy in Step 2.** To isolate the cost of the causal component, we further compare U$^3$CF Step 1-Stage 2 and Step 2. After introducing the causal strategy, the parameter count and peak memory remain unchanged, while the increases in training cost, computational complexity, and inference latency are all limited. This indicates that the causal strategy introduces only limited extra overhead.
>
> **Overall**, although the memory module increases the peak parameter count by 2.8M over D-CGCT, it does not hurt efficiency. U$^3$CF remains faster in both training and inference, likely because it has a simpler network structure and does not adopt the CDAN architecture or the GRL strategy used in many MTDA methods. Moreover, the results show that the memory bank, dual classifiers, and causal strategy do not incur memory or efficiency overhead compared with standard MTDA baselines.
>
> **We will add this analysis in the revised version**.
> Method|Mem Bank|Params (M)|Peak Mem (GB)|Train Time 500iters (s)|FLOPs (G)|Inference Latency (ms/image)|Inference Throughput (img/s)|
> -|-|-|-|-|-|-|-
> D-CGCT (step1)|✗|24.2|13.6|195|263.6|1.01|990.9
> D-CGCT (step2)|✗|26.3|13.6|507|527.9|1.04|965.9
> **D-CGCT (step3)**|✗|24.2|13.6|510|527.6|1.05|954.7
> U$^3$CF (step1-stage1)|✓|29.1|13.1|130|261.8|0.89|1128.6
> U$^3$CF (step1-stage2)|✓|29.1|13.1|275|524.0|0.93|1070.7
> **U$^3$CF (step2)**|✓|29.1|13.1|293|528.1|0.98|1020.4
> >**Q3.** Formatting and Typographical Revisions.
>
> **A3.** We appreciate the reviewer's careful reading and helpful feedback. We will further polish the formatting and typography in the revised version, including details such as the trailing punctuation in some equations.
> >**Q4.** The GNN/GCN nomenclature should be standardized for clarity.
>
> **A4.**
> We appreciate the reviewer's careful reading and helpful feedback. GNN (Graph Neural Network) is a general term, whereas GCN (Graph Convolutional Network) is the specific architecture used in our method. For clarity and consistency, we will standardize the terminology in the revised version and consistently use **GCN** throughout the paper.

---

> > ### Author Rebuttal · Reviewer_2ha6 · 2026-04-02
> >
> > All my concerns have been addressed.

---

> > > ### Author Response · Authors · 2026-04-05
> > >
> > > **Dear Reviewer 2ha6,**
> > >
> > > Thank you very much for your time and thoughtful consideration of our rebuttal.
> > >
> > > We sincerely appreciate your acknowledgment that our responses have addressed your concerns.
> > >
> > > In the revised version, we will revise the manuscript accordingly and incorporate the corresponding clarifications based on your suggestions.
> > >
> > > If you find the clarifications and additional evidence helpful, we would be grateful if they could be considered in your final evaluation.
> > >
> > > Best regards,
> > >
> > > Authors of #10630

---

### Decision · Program_Chairs · 2026-04-30

**Decision:**

Accept (regular)

**Comment:**

**Summary:** This paper studies multi-target domain adaptation through the lens of causality and proposes U3CF, a prototype-driven alignment strategy that progressively updates prototypes by high-confidence target predictions, while the contrastive optimization objective jointly aligns target samples to semantic prototypes and preserves class discrimination.

**Decision:** Overall, the reviewers were positive about this paper and found it could be a solid contribution to the MTDA community. In particular, reviewers appreciated the theoretical analysis, causal perspective for MTDA, explicitly identifying and modeling domain-specific factors as confounders within an SCM, and solid empirical justifications across multiple benchmarks.

Reviewers raised concerns about the computational complexity, training latency, the number of hyperparameters, and the need for several further justifications, including the independence assumptions, the use of a uniform prior, the prototype-based confounder approximation, etc. The authors provided additional analyses and ablation studies, and further clarified the motivation behind the modeling assumptions. These clarifications appear to have addressed the main concerns for most reviewers: Reviewer 2ha6 stated that all concerns were addressed; Reviewer F8Ys also indicated that the questions had been addressed. Reviewers fMWh and 7HsD noted that concerns had been partially addressed while still maintaining a positive assessment.